# Matrix factorisation and the interpretation of geodesic distance

**Nick Whiteley**
University of Bristol
nick.whiteley@bristol.ac.uk

**Annie Gray**
University of Bristol
annie.gray@bristol.ac.uk

**Patrick Rubin-Delanchy**
University of Bristol
patrick.rubin-delanchy@bristol.ac.uk

## Abstract

Given a graph or similarity matrix, we consider the problem of recovering a notion of true distance between the nodes, and so their true positions. We show that this can be accomplished in two steps: matrix factorisation, followed by nonlinear dimension reduction. This combination is effective because the point cloud obtained in the first step lives close to a manifold in which latent distance is encoded as geodesic distance. Hence, a nonlinear dimension reduction tool, approximating geodesic distance, can recover the latent positions, up to a simple transformation. We give a detailed account of the case where spectral embedding is used, followed by Isomap, and provide encouraging experimental evidence for other combinations of techniques.

## 1 Introduction

Assume we observe, or are given as the result of a computational procedure, data in the form of a symmetric matrix $\mathbf{A} \in \mathbb{R}^{n \times n}$ which we relate to unobserved vectors $Z_1, \ldots, Z_n \in \mathcal{Z} \subset \mathbb{R}^d$, where $d \ll n$, by

$$\mathbf{A}^{ij} = f(Z_i, Z_j) + \mathbf{E}^{ij}, \quad 1 \le i \le j \le n, \tag{1}$$

for some real-valued symmetric function $f$ which will be called a kernel, and a matrix of unobserved perturbations, $\mathbf{E} \in \mathbb{R}^{n \times n}$. As illustrative examples, $\mathbf{A}$ could be the observed adjacency matrix of a graph with $n$ vertices, a similarity matrix associated with $n$ items, or some matrix-valued estimator of covariance or correlation between $n$ variates.

Broadly stated, our goal is to recover $Z_1, \ldots, Z_n$ given $\mathbf{A}$, up to identifiability constraints. We seek a method to perform this recovery which can be implemented *in practice* without any knowledge of $f$, nor of any explicit statistical model for $\mathbf{A}$, and to which we can plug in various matrix-types, be it adjacency, similarity, covariance, or other.

At face value, this may seem rather a lot to ask. Without assumptions, neither $f$, the $Z_i$'s, nor even $d$ are identifiable [24]. However, under quite general assumptions, we will show that a practical solution is provided by the key combination of two procedures: i) matrix factorisation of $\mathbf{A}$, such as spectral embedding, followed by ii) nonlinear dimension reduction, such as Isomap [57].

This combination is effective because of the following facts, only the first of which is already known. The matrix factorisation step approximates high-dimensional images of the $Z_i$'s, living near a $d$-dimensional manifold [45]. Under regularity and non-degeneracy assumptions on the kernel, which for our analysis is taken to be positive definite, geodesic distance in this manifold equals Euclidean

35th Conference on Neural Information Processing Systems (NeurIPS 2021).

geodesic distance on $\mathcal{Z}$, up to simple transformations of the coordinates, for example scaling. Thus a dimension reduction technique which approximates in-manifold geodesic distances can be expected to recover the $Z_i$'s, up to a simple transformation. If those assumptions fail, as they must with real data, we may still find that those approximate geodesic distances are useful because they reflect a geometry implicit in the kernel.

There are many works which address recovery of the $Z_i$'s when one *does* consider a particular matrix-type, an explicit statistical model, and/or specific form for $f$. When $\mathbf{A}$ is an adjacency matrix and $f(Z_i, Z_j)$ is the probability of an edge between nodes $i$ and $j$, the construct (1) is a latent position model [22] and various scenarios have been studied: $\mathcal{Z}$ is a sphere and $f(x, y)$ is a function of $\langle x, y \rangle$ [10]; estimation by graph distance when $f(x, y)$ is a function of $\|x - y\|$ [11]; spectral estimation when $f(x, y)$ is a possibly indefinite inner-product [53, 55, 12, 46]; inference under a logistic model via MCMC [22] or variational approximation [48]. The case where $\mathcal{Z}$ is a discrete set of points corresponds to the very widely studied stochastic block model [23], see [5, 33, 44, 53, 8, 14] and references therein; by contrast the methods we propose are directed at the case where $\mathcal{Z}$ is continuous. The opposite problem of estimating $f$, with $Z_i$ unknown but assumed uniform on $[0, 1]$, is known as graphon estimation [19].

Concerning the case when $\mathbf{A}$ is a covariance matrix, Latent Variable Gaussian Process models [31, 30, 59] use $f(Z_i, Z_j)$ to define the population covariance between the $i$th and $j$th variates under a hierarchical model. When $f(x, y) = \langle x, y \rangle$, this reduces to a latent variable optimisation counterpart of Probabilistic PCA [58] and the maximum likelihood estimate of the $Z_i$'s is obtained from the eigendecomposition of the empirical covariance matrix, which in our setting could be $\mathbf{A}$. When $f$ is nonlinear, but fixed e.g. to a Radial Basis Function kernel, gradient and variational techniques are available [31, 30, 59]. See the same references for onward connections to kernel PCA [50].

**Our contribution and precursors.** The general practice of using dimensionality reduction and geodesic distance to extract latent features from data is common in data science and machine-learning. Examples arise in genomics [38], neuroscience [63], speech analysis [28, 21] and cyber-security [9]. What sets our work apart from these contributions is that we establish a new rigorous basis for this practice.

It has been understood for several years that the high-dimensional embedding obtained by matrix factorisation can be related via a feature map to the $Z_i$ of model (1) [26, 15, 34, 56, 62, 32]. However, it was only recently observed that the embedding must therefore concentrate about a low dimensional set, in the Hausdorff sense [45]. Our key mathematical contribution is to describe the topology and geometry of this set, proving it is a topological manifold and establishing how in-manifold geodesic distance is related to geodesic distance in $\mathcal{Z}$. Riemannian geometry underlying kernels was sketched in [7] but without consideration of geodesic distances or rigorous proofs, and not in the context of latent position estimation. The work [13, 61] was an inspiration to us, suggesting Isomap as a tool for analysing spectral embeddings, under a latent structure model in which $\mathcal{Z}$ is a one-dimensional curve in $\mathbb{R}^d$ and $f$ is the inner product. A key feature of our problem setup is that $f$ is unknown, in which case the manifold is not available in closed form and typically lives in an infinite-dimensional space. To our knowledge, we are the first to show why spectral embedding followed by Isomap might recover the true $Z_i$'s, or a useful transformation thereof, in general.

We complement our mathematical results with experimental evidence, obtained from both simulated and real data, suggesting that alternative combinations of matrix-factorisation and dimension-reduction techniques work too. For the former, we consider the popular node2vec [20] algorithm, a likelihood-based approach for graphs said to perform matrix factorisation implicitly [43, 64]. For the latter, we consider the popular t-SNE [37] and UMAP [39] algorithms. As predicted by the theory, a direct low-dimensional matrix factorisation, whether using spectral embedding or node2vec, is less successful.

## 2 Proposed methods and their rationale

### 2.1 Spectral embedding, as estimating $\phi(Z_i)$

In our mathematical setup (precise details come later) the kernel $f$ will be nonnegative definite and $\phi$ will be the associated Mercer feature map. It is well known that each point $\phi(Z_i)$ then lives in an

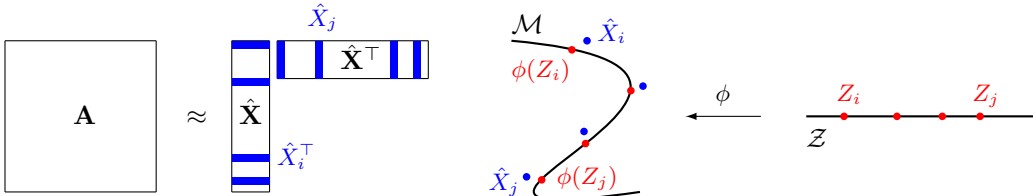

Figure 1: Illustration of theory in the case $d = 1$. Our analysis reveals how geodesic distance, along $\mathcal{M}$, between $\phi(Z_i)$ and $\phi(Z_j)$, is related to geodesic distance, along $\mathcal{Z}$, between $Z_i$ and $Z_j$.

infinite-dimensional Hilbert space, which will be denoted $\ell_2$, and the inner-product $\langle \phi(Z_i), \phi(Z_j) \rangle_2$ in this space equals $f(Z_i, Z_j)$.

**The spectral embedding procedure.** For $p \leq n$, we define the $p$-dimensional spectral embedding of $\mathbf{A}$ to be $\hat{\mathbf{X}} = [\hat{X}_1, \ldots, \hat{X}_n]^\top = \hat{\mathbf{U}}|\hat{\mathbf{S}}|^{1/2} \in \mathbb{R}^{n \times p}$, where $|\hat{\mathbf{S}}| \in \mathbb{R}^{p \times p}$ is a diagonal matrix containing the absolute values of the $p$ largest eigenvalues of $\mathbf{A}$, by magnitude, and $\hat{\mathbf{U}} \in \mathbb{R}^{n \times p}$ is a matrix containing corresponding orthonormal eigenvectors, in the same order. The R packages irlba and RSpectra provide fast solutions which can exploit sparse inputs.

One should think of $\hat{X}_i$ as approximating the vector of first $p$ components of $\phi(Z_i)$, denoted $\phi_p(Z_i)$, up to orthogonal transformation, and this can be formalised to a greater or lesser extent depending on what assumptions are made. There are several situations, e.g. $f$ any polynomial [45], the cosine kernel used in Section 4.1, the degree-corrected [29] or mixed-membership [6] stochastic block model, in which only the first $p_0$ (say) components of $\phi(\cdot)$ are nonzero, where typically $p_0 \geq d$. If, after $n$ reaches $p_0$, we embed into $p = p_0$ dimensions, then with $\| \cdot \|$ denoting the Euclidean norm, we have [18]:

$$\max_{i \in \{1, \ldots, n\}} \|\mathbf{Q}\hat{X}_i - \phi_p(Z_i)\| = O_{\mathbb{P}}\left( \frac{(\log n)^c}{n^{1/2}} \right), \tag{2}$$

for a universal constant $c \geq 1$, orthogonal matrix $\mathbf{Q}$, under regularity assumptions on the $Z_i$'s, $f$ and $\mathbf{E}$ (that $Z_i$ are i.i.d., $f(Z_i, Z_j)$ has finite expectation, and the perturbations $\mathbf{E}^{ij}$ are independent and centered with exponential tails). This encompasses the case where $\mathbf{A}$ is binary, for example a graph adjacency matrix [36, 46]. Similar results are available in the cases where $\mathbf{A}$ is a Laplacian [46, 40] or covariance matrix [17]. The methods of this paper are based, in practice, on the distances $\|\hat{X}_i - \hat{X}_j\|$, which are invariant to orthogonal transformation and so for the purposes of validating $\|\hat{X}_i - \hat{X}_j\| \approx \|\phi_p(Z_i) - \phi_p(Z_j)\|$ the presence of $\mathbf{Q}$ in (2) is immaterial.

For $\hat{X}_i$ to converge to $\phi(Z_i)$ more generally, we must let its dimension $p$ grow with $n$ and, at least given the present state of literature, accept weaker consistency results, for example, convergence in Wasserstein distance between $\mathbf{Q}\hat{X}_1, \ldots \mathbf{Q}\hat{X}_n$ and $\phi_p(Z_1), \ldots, \phi_p(Z_n)$ [32]. Uniform consistency results, in the style of (2), are also available for indefinite [46], bipartite, and directed graphs [25]. These are left for future work because of the complications of $\mathbf{Q}$ no longer being orthogonal.

**Rank selection.** In real data, where $n$ is typically fixed, there is no 'best' way of selecting $p$, as discussed for example in [42]. The method of [65], based on profile-likelihood, provides a popular, practical choice, taking as input the spectrum of $\mathbf{A}$, and is implemented in the R package igraph.

## 2.2 Isomap, as estimating $Z_i$

We propose to recover $Z_1, \ldots, Z_n$ through the following procedure.

---
**Algorithm 1** Isomap procedure
---
    **input** $p$-dimensional points $\hat{X}_1, \ldots, \hat{X}_n$
1: Compute the neighbourhood graph of radius $\epsilon$: a weighted graph connecting $i$ and $j$, with weight $\|\hat{X}_i - \hat{X}_j\|$, if $\|\hat{X}_i - \hat{X}_j\| \leq \epsilon$
2: Compute the matrix of shortest paths on the neighbourhood graph, $\hat{\mathbf{D}}_{\mathcal{M}} \in \mathbb{R}^{n \times n}$
3: Apply classical multidimensional scaling (CMDS) to $\hat{\mathbf{D}}_{\mathcal{M}}$ into $\mathbb{R}^d$
    **return** $d$-dimensional points $\hat{Z}_1, \ldots, \hat{Z}_n$
---

Theorem 1 below establishes, under general assumptions, that $\mathcal{M} := \phi(\mathcal{Z})$ is a topological manifold of Hausdorff dimension exactly that of $\mathcal{Z}$ (as opposed to an upper bound [45]). It also proposes a 'change of metric', often trivial, under which a path on $\mathcal{Z}$ and its image on $\mathcal{M}$ have the same length.

This result explains how $\hat{Z}_1, \ldots, \hat{Z}_n$ can estimate $Z_1, \ldots, Z_n$. First, think of a path from $\hat{X}_i$ to $\hat{X}_j$ on the neighbourhood graph as a noisy, discrete version of some corresponding continuous path from $\phi(Z_i)$ to $\phi(Z_j)$ on $\mathcal{M}$. The length of the first (the sum of the weights of the edges) is approximately equal to the length of the second if measured in the standard metric: an infinitesmal step from $x$ to $x + \mathrm{d}x$ has length $\langle \mathrm{d}x, \mathrm{d}x \rangle_2^{1/2}$. By inversion of $\phi$ we can trace a third path, taking us from $Z_i$ to $Z_j$ on $\mathcal{Z}$. But to make its length agree with the first two, we must pick a non-standard metric: an infinitesmal step from $z$ to $z + \mathrm{d}z$ must be regarded to have length $\langle \mathrm{d}z, \mathbf{H}_z \mathrm{d}z \rangle^{1/2} = \sqrt{\mathrm{d}z^\top \mathbf{H}_z \mathrm{d}z}$, where

$$\mathbf{H}_z^{ij} := \left. \frac{\partial^2 f}{\partial x^{(i)} \partial y^{(j)}} \right|_{(z,z)}, \quad i, j \in \{1, \ldots, d\}.$$

The exciting news for practical purposes is that, through elementary calculus, we might establish that $\mathbf{H}_z$ is constant in $z$ (e.g. if $f$ is translation-invariant) or even proportional to the identity (e.g. if $f$ is just a function of Euclidean distance). In the latter case, if $\mathcal{Z}$ is convex, it is not too difficult to see that the length of the *shortest* path, from $\phi(Z_i)$ to $\phi(Z_j)$ on $\mathcal{M}$, must be proportional to the Euclidean distance between $Z_i$ and $Z_j$. Thus, the matrix of shortest paths obtained by Isomap (Step 2) approximates the matrix of Euclidean distances between the $Z_i$ (up to scaling), from which the $Z_i$ themselves can be recovered (Step 3), up to scaling, rotation, and translation. Of course, we have taken a few liberties in this argument, such as assuming $\phi$ to be invertible and $\mathrm{d}z^\top \mathbf{H}_z \mathrm{d}z$ to be positive. We will address these rigorously in the next section.

**Dimension and radius selection.** For visualisation in our examples we will pick $d = 1$ or $d = 2$. For other applications, $d$ can be estimated as the approximate rank of $\hat{\mathbf{D}}_{\mathcal{M}}$ after double-centering [60], e.g. via [65], again. In practice, we suggest picking $\epsilon$ just large enough for the neighbourhood graph to be connected or, if the data have outliers, a fixed quantile (such as 5%) of $\hat{\mathbf{D}}_{\mathcal{M}}$, removing nodes outside the largest connected component. The same recommendations apply if the $k$-nearest neighbour graph is used instead of the $\epsilon$-neighbourhood graph.

## 3 Theory

### 3.1 Setup and assumptions

The usual inner-product and Euclidean norm on $\mathbb{R}^d$ are denoted $\langle \cdot, \cdot \rangle$ and $\| \cdot \|$. $\ell_2$ is the set of $x = [x^{(1)} \, x^{(2)} \, \cdots]^\top \in \mathbb{R}^{\mathbb{N}}$ such that $\|x\|_2 := (\sum_{k=1}^\infty |x^{(k)}|^2)^{1/2} < \infty$. For $x, y \in \ell_2$, $\langle x, y \rangle_2 := \sum_{k=1}^\infty x^{(k)} y^{(k)}$. With $d \geq 1$, let $\mathcal{Z}$ be a compact subset of $\mathbb{R}^d$, and let $\widetilde{\mathcal{Z}}$ be a closed ball in $\mathbb{R}^d$, centered at the origin, such that $\mathcal{Z} \subset \widetilde{\mathcal{Z}}$. Let $f : \widetilde{\mathcal{Z}} \times \widetilde{\mathcal{Z}} \to \mathbb{R}$ be a symmetric, continuous, nonnegative-definite function.

By Mercer's Theorem, e.g., [51, Thm 4.49], there exist nonnegative real numbers $(\lambda_k)_{k \geq 1}$ and functions $(u_k)_{k \geq 1}$, with each $u_k : \widetilde{\mathcal{Z}} \to \mathbb{R}$, which are orthonormal with respect to the inner-product $(u_j, u_k) \mapsto \int_{\widetilde{\mathcal{Z}}} u_j(x) u_k(x) \mathrm{d}x$ and such that

$$f(x, y) = \sum_{k=1}^\infty \lambda_k u_k(x) u_k(y), \quad x, y \in \widetilde{\mathcal{Z}}, \tag{3}$$

where the convergence is absolute and uniform. For $x \in \mathcal{Z}$ let $\phi(x) := [\lambda_1^{1/2} u_1(x) \ \lambda_2^{1/2} u_2(x) \ \cdots]^\top \in \mathbb{R}^\mathbb{N}$. The image of $\mathcal{Z}$ by $\phi$ is denoted $\mathcal{M}$. Observe from (3) that for any $x, y \in \mathcal{Z}$, $f(x, y) = \langle \phi(x), \phi(y) \rangle_2$, $\|\phi(x)\|_2^2 = f(x, x)$, and the latter is finite for any $x \in \mathcal{Z}$ since $f$ is continuous and $\mathcal{Z}$ is compact, hence $\mathcal{M} \subset \ell_2$.

The following definitions are standard in metric geometry [16]. For any $a, b \in \ell_2$, a *path* in $\mathcal{M}$ with end-points $a, b$ is a continuous function $\gamma : [0, 1] \to \mathcal{M}$ such that $\gamma_0 = a$ and $\gamma_1 = b$. With $n \geq 1$, a non-decreasing sequence $t_0, \ldots, t_n$ such that $t_0 = 0$ and $t_n = 1$, is called a *partition*. Given a path $\gamma$ and a partition $\mathcal{P} = (t_0, \ldots, t_n)$, define $\chi(\gamma, \mathcal{P}) := \sum_{k=1}^n \|\gamma_{t_k} - \gamma_{t_{k-1}}\|_2$. The *length* of $\gamma$ (w.r.t. $\|\cdot\|_2$) is $l(\gamma) := \sup_\mathcal{P} \chi(\gamma, \mathcal{P})$, where the supremum is over all possible partitions. The geodesic distance in $\mathcal{M}$ between $a$ and $b$ is defined to be the infimum of $l(\gamma)$ over all paths $\gamma$ in $\mathcal{M}$ with end-points $a, b$. $\mathbf{D}_\mathcal{M}^{ij}$ denotes geodesic distance in $\mathcal{M}$ between $\phi(Z_i)$ and $\phi(Z_j)$. Similarly a path $\eta$ in $\mathcal{Z}$ with end-points $x, y$ is a continuous function $\eta : [0, 1] \to \mathcal{Z}$ such that $\eta_0 = x, \eta_1 = y$, and with $\chi(\eta, \mathcal{P}) := \sum_{k=1}^n \|\eta_{t_k} - \eta_{t_{k-1}}\|$ the length of $\eta$ (w.r.t. $\|\cdot\|$) is $l(\eta) := \sup_\mathcal{P} \chi(\eta, \mathcal{P})$. $\mathbf{D}_\mathcal{Z}^{ij}$ denotes geodesic distance in $\mathcal{Z}$ between $Z_i$ and $Z_j$. If $\mathcal{Z}$ is convex, $\mathbf{D}_\mathcal{Z}^{ij} = \|Z_i - Z_j\|$.

**Assumption 1.** *For all $x, y \in \mathcal{Z}$ with $x \neq y$, there exists $a \in \mathcal{Z}$ such that $f(x, a) \neq f(y, a)$.*

**Assumption 2.** *$f$ is $C^2$ on $\widetilde{\mathcal{Z}}$ and for every $z \in \mathcal{Z}$, the matrix $\mathbf{H}_z$ is positive definite.*

Assumption 1 is used to show $\phi$ is injective. Assumption 2 has various implications, loosely speaking it ensures a non-degenerate relationship between path-length in $\mathcal{M}$ and path-length in $\mathcal{Z}$. Concerning our general setup, if $f$ is not required to be nonnegative definite, but is still symmetric, then a representation formula like (3) is available under e.g. trace-class assumptions [45]. It is of interest to generalise our results to that scenario but technical complications are involved, so we leave it for future research.

## 3.2 Path-lengths in $\mathcal{M}$ and in $\mathcal{Z}$

**Theorem 1.** *Let assumptions 1 and 2 hold. Then $\phi$ is a bi-Lipschitz homeomorphism between $\mathcal{Z}$ and $\mathcal{M}$. Let $a, b$ be any two points in $\mathcal{M}$ such that there exists a path in $\mathcal{M}$ with end-points $a, b$ of finite length. Let $\gamma$ be any such path and define $\eta : [0, 1] \to \mathcal{Z}$ by $\eta_t := \phi^{-1}(\gamma_t)$. Then $\eta$ is a path in $\mathcal{Z}$ with $l(\eta) < \infty$. For any $\epsilon > 0$ there exists a partition $\mathcal{P}_\epsilon$ such that for any partition $\mathcal{P} = (t_0, \ldots, t_n)$ satisfying $\mathcal{P}_\epsilon \subseteq \mathcal{P}$,*

$$\left| l(\gamma) - \sum_{k=1}^n \left\langle \eta_{t_k} - \eta_{t_{k-1}}, \mathbf{H}_{\eta_{t_{k-1}}} (\eta_{t_k} - \eta_{t_{k-1}}) \right\rangle^{1/2} \right| \leq \epsilon. \tag{4}$$

*If $\gamma$ and $\eta$ are continuously differentiable, the following two equalities hold:*

$$l(\gamma) = \int_0^1 \|\dot{\gamma}_t\|_2 \mathrm{d}t = \int_0^1 \langle \dot{\eta}_t, \mathbf{H}_{\eta_t} \dot{\eta}_t \rangle^{1/2} \, \mathrm{d}t. \tag{5}$$

The proof is in the appendix. In the terminology of differential geometry, the collection of inner-products $(x, y) \mapsto \langle x, \mathbf{H}_z y \rangle$, indexed by $z \in \mathcal{Z}$, constitute a *Riemannian metric* on $\mathcal{Z}$, see e.g. [41, Ch.1 and p.121] for background, and the right hand side of (5) is the length of the path $\eta$ in $\mathcal{Z}$ with respect to this Riemannian metric. The theorem tells us that finding geodesic distance in $\mathcal{M}$, i.e. minimising $l(\gamma)$ with respect to $\gamma$ for fixed end-points say $a = \phi(Z_i), b = \phi(Z_j)$, is equivalent to minimizing path-length in $\mathcal{Z}$ between end-points $\phi^{-1}(a) = Z_i, \phi^{-1}(b) = Z_j$ under the Riemannian metric. In section 3.3 we will show how, for various classes of kernels, this minimisation can be performed in closed form leading to explicit relationships between $\mathbf{D}_\mathcal{M}^{ij}$ and $\mathbf{D}_\mathcal{Z}^{ij}$. In Section 3.3 we focus on (5) rather than (4) only for ease of exposition, it is shown in the appendix that exactly the same conclusions can be derived directly from (4). To avoid repetition Assumption 1 will be taken to hold throughout section 3.3 without further mention.

## 3.3 Geodesic distance in $\mathcal{M}$ and in $\mathcal{Z}$

**Translation invariant kernels.** Suppose $\mathcal{Z} \subset \mathbb{R}^d$ is compact and convex, and $f(x, y) = g(x - y)$ where $g : \mathbb{R}^d \to \mathbb{R}$ is $C^2$. In this situation $\mathbf{H}_z$ is constant in $z$, and equal to the Hessian of $-g$ evaluated at the origin. The positive-definite part of Assumption 2 is therefore satisfied if and only if

$g$ has a local maximum at the origin. We now use Theorem 1 to find geodesic distance in $\mathcal{M}$ between generic points $a, b \in \mathcal{M}$ for this class of translation-invariant kernels. To do this we obtain a lower bound on $l(\gamma)$ over all paths $\gamma$ in $\mathcal{M}$ with end-points $a, b$, then show there exists such a path whose length achieves this lower bound.

Let $\mathbf{G} = \mathbf{V}\mathbf{\Lambda}^{1/2}$ where $\mathbf{V}\mathbf{\Lambda}\mathbf{V}^\top$ is the eigendecomposition of the Hessian of $-g$ evaluated at the origin, so $\mathbf{H}_z = \mathbf{G}\mathbf{G}^\top$ for all $z$. Then from (5), for a generic path $\gamma$ in $\mathcal{M}$ with end-points $a, b$,

$$l(\gamma) = \int_0^1 \langle \dot{\eta}_t, \mathbf{H}_{\eta_t} \dot{\eta}_t \rangle^{1/2}\, \mathrm{d}t = \int_0^1 \langle \dot{\eta}_t, \mathbf{G}\mathbf{G}^\top \dot{\eta}_t \rangle^{1/2}\, \mathrm{d}t = \int_0^1 \|\mathbf{G}^\top \dot{\eta}_t\| \mathrm{d}t \tag{6}$$

$$\geq \left\| \int_0^1 \mathbf{G}^\top \dot{\eta}_t \mathrm{d}t \right\| = \left\| \mathbf{G}^\top \int_0^1 \dot{\eta}_t \mathrm{d}t \right\| = \|\mathbf{G}^\top (\eta_1 - \eta_0)\| = \|\mathbf{G}^\top [\phi^{-1}(b) - \phi^{-1}(a)]\|, \tag{7}$$

where the inequality is due to the triangle inequality for the $\|\cdot\|$ norm. The right-most term in (7) is independent of $\gamma$, other than through the end-points $a, b$. To see there exists a path whose length equals this lower bound, take

$$\tilde{\eta}_t \coloneqq \phi^{-1}(a) + t[\phi^{-1}(b) - \phi^{-1}(a)], \quad t \in [0, 1], \tag{8}$$

which is well-defined as a path in $\mathcal{Z}$, since for this class of kernels $\mathcal{Z}$ is assumed convex. With $\tilde{\gamma}_t \coloneqq \phi(\tilde{\eta}_t)$, $\tilde{\gamma}$ is clearly a path in $\mathcal{M}$. Differentiating $\tilde{\eta}_t$ w.r.t. $t$ and substituting into (6) shows $l(\tilde{\gamma}) = \|\mathbf{G}^\top [\phi^{-1}(b) - \phi^{-1}(a)]\|$, as required. The following proposition summarises our conclusions in the case that $a = \phi(Z_i)$ and $b = \phi(Z_j)$, for any $Z_i, Z_j$.

**Proposition 1.** *If $\mathcal{Z}$ is compact and convex, and $f(x, y) = g(x - y)$ where $g : \mathbb{R}^d \to \mathbb{R}$ is $C^2$ and has a local maximum at the origin, then $\mathbf{D}_{\mathcal{M}}^{ij}$ is equal to Euclidean distance between $Z_i$ and $Z_j$, up to their linear transformation by $\mathbf{G}^\top$. In the particular case where $g(x - y) = h(\|x - y\|^2)$ with $h'(0) < 0$, we have $\mathbf{D}_{\mathcal{M}}^{ij} \propto \mathbf{D}_{\mathcal{Z}}^{ij} = \|Z_i - Z_j\|$.*

**Inner-product kernels.** Suppose $\mathcal{Z} = \{x \in \mathbb{R}^d : \|x\| = 1\}$ and $f(x, y) = g(\langle x, y \rangle)$, where $g : \mathbb{R} \to \mathbb{R}$ is $C^2$ and such that $g'(1) > 0$. In this case $\mathbf{H}_z = g'(1)\mathbf{I} + g''(1)zz^\top$, and by a result of [27], any kernel of the form $f(x, y) = g(\langle x, y \rangle)$ on the sphere is positive-definite iif $g(x) = \sum_{n=0}^\infty a_n x^n$ for some nonnegative $(a_n)_{n \geq 0}$, implying $g''(1) \geq 0$. Assumption 2 then holds. To derive the geodesic distance in $\mathcal{M}$, first write out (5) for a generic path $\gamma$ in $\mathcal{M}$ with end-points $a, b$:

$$l(\gamma) = \int_0^1 \langle \dot{\eta}_t, \mathbf{H}_{\eta_t} \dot{\eta}_t \rangle^{1/2}\, \mathrm{d}t = \int_0^1 \left( g'(1)\|\dot{\eta}_t\|^2 + g''(1)\, |\langle \eta_t, \dot{\eta}_t \rangle|^2 \right)^{1/2}\, \mathrm{d}t. \tag{9}$$

Since $\mathcal{Z}$ is a radius-1 sphere centered at the origin we must have $\|\eta_t\| = 1$ for all $t$, so $0 = \frac{1}{2}\frac{\mathrm{d}}{\mathrm{d}t}\|\eta_t\|^2 = \frac{1}{2}\frac{\mathrm{d}}{\mathrm{d}t}\langle \eta_t, \eta_t \rangle = \langle \eta_t, \dot{\eta}_t \rangle$. Therefore the r.h.s. of (9) is in fact equal to $g'(1)^{1/2} \int_0^1 \|\dot{\eta}_t\| \mathrm{d}t$. This is minimised over all possible paths $\eta$ in $\mathcal{Z}$ with end-points $\phi^{-1}(a), \phi^{-1}(b)$ when $\eta$ is a shortest (with respect to Euclidean distance) circular arc in $\mathcal{Z}$, in which case $\int_0^1 \|\dot{\eta}_t\| \mathrm{d}t = \arccos \langle \phi^{-1}(a), \phi^{-1}(b) \rangle$. Thus we have:

**Proposition 2.** *If $\mathcal{Z} = \{x \in \mathbb{R}^d : \|x\| = 1\}$ and $f(x, y) = g(\langle x, y \rangle)$ where $g : \mathbb{R} \to \mathbb{R}$ is $C^2$ and such that $g'(1) > 0$, then $\mathbf{D}_{\mathcal{M}}^{ij} = g'(1)^{1/2}\mathbf{D}_{\mathcal{Z}}^{ij} = g'(1)^{1/2} \arccos \langle Z_i, Z_j \rangle$.*

**Additive kernels.** Suppose $\mathcal{Z}$ is the Cartesian product of intervals $\mathcal{Z}_i = [c_i^-, c_i^+]$, $i = 1, \ldots, d$ and $f(x, y) = \sum_{i=1}^d \alpha_i f_i(x^{(i)}, y^{(i)})$, $x = [x^{(1)} \cdots x^{(d)}]^\top$, where each $\alpha_i > 0$ and each $f_i$ is positive-definite and $C^2$. For this class of kernels $\mathbf{H}_z$ is diagonal with $\mathbf{H}_z^{ii} = \alpha_i \left. \frac{\partial^2 f_i}{\partial x^{(i)} \partial y^{(i)}} \right|_{(z^{(i)}, z^{(i)})}$ where $z^{(i)}$ is the $i$th element of the vector $z$. The positive-definite part of Assumption 2 thus holds if these diagonal elements are strictly positive for all $z \in \mathcal{Z}$.

Let us introduce the vector of monotonically increasing transformations:

$$\psi(z) \coloneqq [\psi_1(z^{(1)}) \cdots \psi_d(z^{(d)})]^\top, \qquad \psi_i(z^{(i)}) \coloneqq \alpha_i^{1/2} \int_{c_i^-}^{z^{(i)}} \left. \frac{\partial^2 f_i}{\partial x^{(i)} \partial y^{(i)}} \right|_{(\xi, \xi)}^{1/2}\, \mathrm{d}\xi, \tag{10}$$

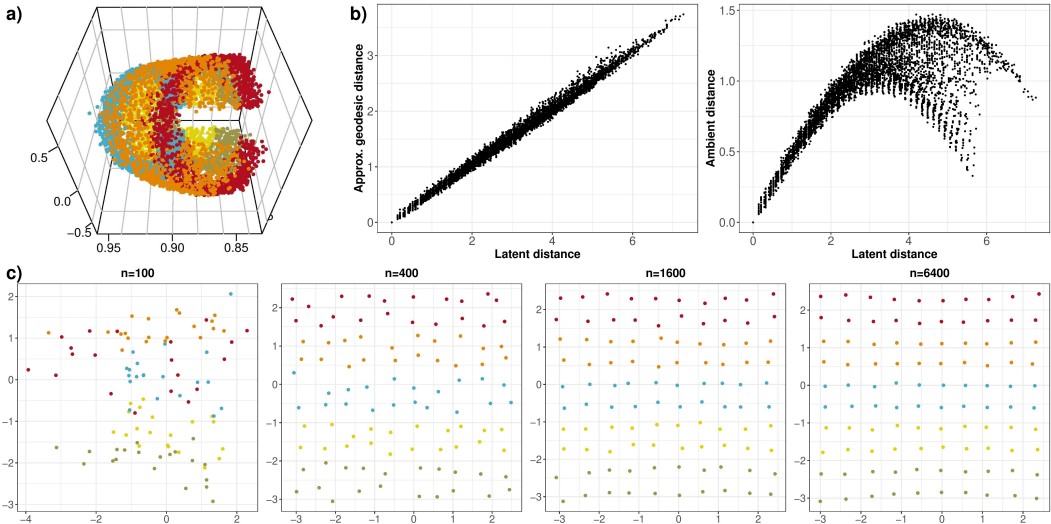

Figure 2: Simulated data example. a) Spectral embedding (first 3 dimensions), b) comparison of latent, approximate geodesic, and ambient distance and c) latent position recovery in the dense regime by spectral embedding followed by Isomap for increasing $n$. To aid visualisation, all plots in c) display a subset of 100 estimated positions corresponding to true positions on a sub-grid which is common across $n$. Estimated positions are coloured according to their true $y$-coordinate.

and write the vector of corresponding inverse transformations $\psi^{-1}$. For a generic path $\gamma$ in $\mathcal{M}$ with end-points $a, b$ and as usual $\eta_t := \phi^{-1}(\gamma_t)$, define $\zeta_t := \psi(\eta_t)$. Then:

$$l(\gamma) = \int_0^1 \langle \dot{\eta}_t, \mathbf{H}_{\eta_t} \dot{\eta}_t \rangle^{1/2} \, \mathrm{d}t = \int_0^1 \|\dot{\zeta}_t\| \mathrm{d}t \tag{11}$$

$$\geq \left\| \int_0^1 \dot{\zeta}_t \, \mathrm{d}t \right\| = \|\zeta_1 - \zeta_0\| = \|\psi \circ \phi^{-1}(b) - \psi \circ \phi^{-1}(a)\|, \tag{12}$$

where the second and last equalities hold due to the definition of $\zeta_t$, and $\circ$ denotes elementwise composition. The quantity $\|\psi \circ \phi^{-1}(b) - \psi \circ \phi^{-1}(a)\|$ is thus a lower bound on path-length $l(\gamma)$ for any path $\gamma$ in $\mathcal{M}$ with end-points $a, b$. To see that there exists a path whose length achieves this lower bound, and hence that it is the geodesic distance in $\mathcal{M}$ between $a, b$, define $\tilde{\gamma}_t := \phi \circ \psi^{-1}(\tilde{\zeta}_t)$ where $\tilde{\zeta}_t := \psi \circ \phi^{-1}(a) + t[\psi \circ \phi^{-1}(b) - \psi \circ \phi^{-1}(a)]$. Differentiating $\tilde{\zeta}_t$ w.r.t. $t$ and substituting into (11) yields $l(\tilde{\gamma}) = \|\psi \circ \phi^{-1}(b) - \psi \circ \phi^{-1}(a)\|$, as required. Our conclusions in the case that $a = \phi(Z_i)$ and $b = \phi(Z_j)$ are summarised by:

**Proposition 3.** *If $\mathcal{Z}$ is the Cartesian product of $d$ intervals $\mathcal{Z}_i \subset \mathbb{R}$, $i = 1, \ldots, d$, and $f(x, y) = \sum_{i=1}^d \alpha_i f_i(x^{(i)}, y^{(i)})$ where for each $i$, $\alpha_i > 0$, $f_i$ is positive definite and $C^2$, and $\left. \frac{\partial^2 f_i}{\partial x^{(i)} \partial y^{(i)}} \right|_{(z,z)} > 0$ for all $z \in \mathcal{Z}_i$, then $\mathbf{D}_{\mathcal{M}}^{ij}$ is equal to the Euclidean distance between $Z_i$ and $Z_j$ up to their coordinatewise-monotone transformation by $\psi$. If $\alpha_i \left. \frac{\partial^2 f_i}{\partial x^{(i)} \partial y^{(i)}} \right|_{(z,z)}$ is a positive constant over all $z \in \mathcal{Z}_i$, and over all $i = 1, \ldots, d$, then $\mathbf{D}_{\mathcal{M}}^{ij} \propto \mathbf{D}_{\mathcal{Z}}^{ij} = \|Z_i - Z_j\|$.*

## 4 Experiments

### 4.1 Simulated data: latent position network model

This section shows the theory at work in a pedagogical example, where $\mathbf{A}$ is an adjacency matrix. Consider an undirected random graph following a latent position model with kernel $f(x, y) = \rho_n \{\cos(x^{(1)} - y^{(1)}) + \cos(x^{(2)} - y^{(2)}) + 2\}/4$, operating on $\mathbb{R}^2 \times \mathbb{R}^2$. Here, the sequence $\rho_n$ is a sparsity factor which will either be constant, $\rho_n = 1$, or shrinking to zero sufficiently slowly,

reflecting dense (degree grows linearly) and sparse (degree grows sublinearly) regimes respectively. The kernel is clearly translation-invariant, satisfies Assumptions 1 and 2 and has finite rank, $p = 5$. The true latent positions $Z_i \in \mathbb{R}^2$ are deterministic and equally spaced points on a grid over the region $\mathcal{Z} = [-\pi + 0.25, \pi - 0.25] \times [-\pi + 0.25, \pi - 0.25]$, this range chosen to give valid probabilities and an interesting bottleneck in the 2-dimensional manifold $\mathcal{M}$. From Proposition 1, the geodesic distance between $\phi(Z_i)$ and $\phi(Z_j)$ on $\mathcal{M}$ is equal to the Euclidean distance between $Z_i$ and $Z_j$, up to scaling, specifically $\mathbf{D}_{\mathcal{M}}^{ij} = \rho_n \mathbf{D}_{\mathcal{Z}}^{ij}/2$.

Focusing first on the dense regime, for each $n = 100, 400, 1600, 6400$, we simulate a graph, and its spectral embedding $\hat{X}_1, \ldots, \hat{X}_n$ into $p = 5$ dimensions. The first three dimensions of this embedding are shown in Figure 2a), and we see that the points gather about a two-dimensional, curved, manifold. To approximate geodesic distance, we compute a neighbourhood graph of radius $\epsilon$ from $\hat{X}_1, \ldots, \hat{X}_n$, choosing $\epsilon$ as small as possible subject to maintaining a connected graph. Figure 2b) shows that approximated geodesic distances roughly agree with the true Euclidean distances in latent space $\mathcal{Z}$ (up to the predicted scaling of 1/2), whereas there is significant distortion between ambient and latent distance ($\|\hat{X}_i - \hat{X}_j\|$ versus $\|Z_i - Z_j\|$). Finally, we recover the estimated latent positions in $\mathbb{R}^2$ using Isomap, which we align with the original by Procrustes (orthogonal transformation with scaling), in Figure 2c). As the theory predicts, the recovery error vanishes as $n$ increases.

In the appendix, we perform the same experiment in a sparse regime, $n\rho_n = \omega\{\log^4 n\}$, chosen to ensure the spectral embedding is still consistent, and the recovery error still shrinks, but more slowly. We also implement other related approaches: UMAP, t-SNE applied to spectral embedding, node2vec directly into two dimensions and node2vec in five dimensions followed by Isomap (Figure 6). We use default configurations for all other hyperparameters. Together, the results support the central recommendation of this paper: to use matrix factorisation (e.g. spectral embedding, node2vec), followed by nonlinear dimension reduction (e.g. Isomap, UMAP, t-SNE).

## 4.2 Real data: the global flight network

For this example, we use a publically available, clean version [52] (with license details therein) of the crowdsourced OpenSky dataset. Six undirected graphs are extracted, for the months November 2019 to April 2020, connecting any two airports in the world for which at least a single flight was recorded ($n \approx 10,000$ airports in each). For each graph the adjacency matrix $\mathbf{A}$ is spectrally embedded into $p = 10$ dimensions, degree-corrected by spherical projection [35, 33, 49], after which we apply Isomap with $\epsilon$ chosen as the 5% distance quantile.

The dimension $p$ was chosen as a loose upper bound on the dimension estimate returned by the method of [65] on any of the individual graphs, as to facilitate comparison we prefer to use the same dimension throughout (e.g. to avoid artificial differences in variance) and so follow the recommendation of [12] to err on the large side (experiments with different choices of $p$ are in the appendix). In the degree-correction step we aim to remove network effects related to "airport popularity" (e.g. population, economic and airport size), after which geographic distance might be expected to be the principal factor deciding whether there should be a flight between two airports. Therefore, after applying Isomap, we might hope that $\hat{Z}_i$ could estimate the geographical location of airport $i$. The experiments take a few hours on a standard desktop (and the same is a loose upper bound on compute time for the other experiments in the paper).

The embedding for January is shown in Figures 3a-b), before and after Isomap. Both recover real geographic features, notably the relative positioning of London, Paris, Madrid, Lisbon, Rome. However, the embedding of the US is warped, and only after using Isomap do we see New York and Los Angeles correctly positioned at opposite extremes of the country. As further confirmation, Figures 3c-d) show the results restricted to North America, with the points coloured by latitude and longitude. For this continent, the accuracy of longitude recovery is particularly striking.

In line with the recommendation to overestimate $p$ [12], one obtains a similar figure using $p = 20$ before applying Isomap (Figure 10, appendix), whereas the corresponding figure with $p = 2$ is too poor to show.

In the appendix, the embeddings of all six months are shown, aligned using two-dimensional Procrustes, showing an important structural change in April. A statistical analysis of inter-continental

geodesic distances suggests the change reflects severe flight restrictions in South America and Africa, at the beginning COVID-19 pandemic (the 11th March 2020 according to the WHO).

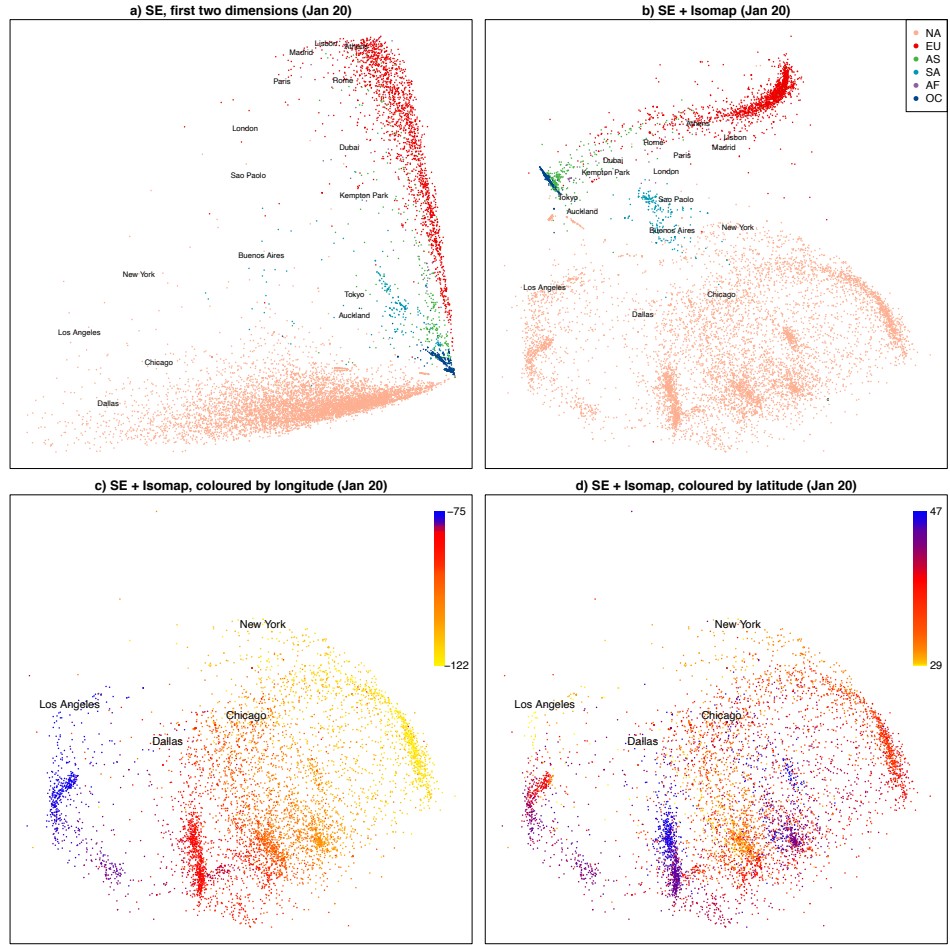

Figure 3: Visualisation of the global flight network, January 2020. a) Spectral embedding, b) spectral embedding followed by Isomap, colours indicating continents (NA = North America, EU = Europe, AS = Asia, AF = Africa, OC = Oceania), c-d) the same restricted to North America, showing only the airports within the continent's 5%-95% inter-quantile range of longitude and latitude, colours indicating the longitude (c) and latitude (d) of the airports.

## 4.3   Real data: correlations between daily temperatures around the globe

In this example $\mathbf{A}$ is a correlation matrix and we demonstrate a simple model-checking diagnostic informed by our theory. The raw data consist of average temperatures over each day, for several thousand days, recorded in cities around the globe. These data are open source, originate from Berkeley Earth [1] and the particular data set analyzed is from [3]. We used open source latitude and longitude data from [4]. See those references for license details. Removing cities and dates for which data are missing yields a temperature time-series of $1450$ common days for each of the $n = 2211$ cities. $\mathbf{A}$ is the matrix of Pearson correlation coefficients between the $n$ time-series.

Figure 4b) shows the spectral embedding of $\mathbf{A}$, with $p = 2$ and points coloured by the latitude of the corresponding cities. Two visual features are striking: the concentration of points around a curved manifold, and a correspondence between latitude and location on the manifold. Figure 4c) shows latitude against estimated latent positions from Isomap using the $k$-nearest neighbour graph with $k = 200$, which is roughly $10\%$ of $n$, and with $d = 1$. A clear monotone relationship appears. Our theoretical results can explain this phenomenon. Notice that when $d = 1$, the additive structure of the kernel in Proposition 3 disappears. Hence Proposition 3 shows that in the case $d = 1$, for *any* kernel

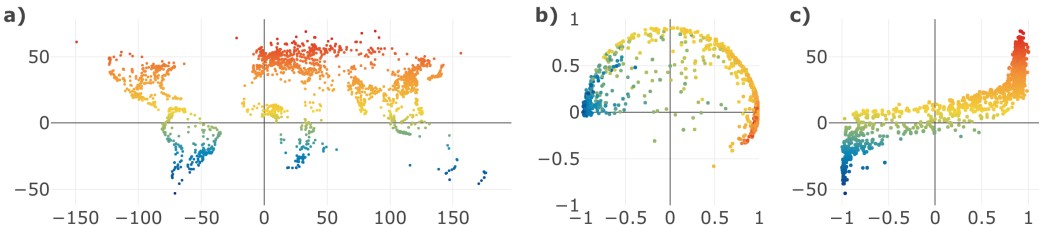

Figure 4: Temperature correlation example. a) Locations in degrees latitude and longitude of cities where temperatures were recorded, b) spectral embedding with $p = 2$, c) latitude of each city (vertical axis) against estimated latent position (horizontal axis) when $d = 1$. In all plots, points are coloured by latitude of the corresponding cities.

(meeting the basic requirements of section 3.1, including Assumptions 1 and 2 of course) $\mathbf{D}_{\mathcal{M}}^{ij}$ is equal to Euclidean distance between $Z_i$ and $Z_j$ up to their monotone transformation by $\psi$ defined in (10). In turn, this implies that if $\mathbf{A}$ did actually follow the model (1) for some $f$ and some "true" latent positions $Z_1, \ldots, Z_n$ to which we had access, we should observe a monotone relationship between those true latent positions and the estimated positions $\hat{Z}_1, \ldots, \hat{Z}_n$ from Isomap.

The empirical monotonicity in Figure 4c) thus can be interpreted as indicating latitudes of the cities are plausible as "true" latent positions underlying the correlation matrix $\mathbf{A}$, without us having to specify anything in practice about $f$. In further analysis (appendix) no such empirical monotone relationship is found between longitude and estimated latent position; this indicates longitude does *not* influence the correlations captured in $\mathbf{A}$. A possible explanation is the daily averaging of temperatures in the underlying data: correlations between these average temperatures may be insensitive to longitude due to the rotation of the earth.

## 5   Conclusion

Our research shows how matrix-factorisation and dimension-reduction can work together to reveal the true positions of objects based only on pairwise similarities such as connections or correlations. For the sake of exposition and reproducibility, we have used public datasets which can be interpreted without specialist domain knowledge, but the methods are potentially relevant to any scientific application involving large similarity matrices. Thinking about societal impact, our results highlight the depth of information in principle available about individuals, given network metadata, and we hope to raise awareness of these potential issues of privacy.

Concerning the limitations of the methods we have discussed, the bound in (2) indicates that $n$ should be "large" in order for $\mathbf{Q}\hat{X}_i$ to approximate $\phi_p(Z_i)$ well. $n$ also has an impact on the performance of Isomap: heuristically one needs a high density and a large number of points on or near the manifold to get a good estimate of the geodesic distance. Thus the methods we propose are likely to perform poorly when $n$ is small, corresponding e.g. to a graph with a small number of vertices.

On the other hand, in applications involving large networks or matrices (e.g. cyber-security, recommender systems), the data encountered are often sparse. This is good news for computational feasibility but bad news for statistical accuracy. In particular, for a graph, (2) can only be expected to hold under logarithmically growing degree, the information-theoretic limit below which no algorithm can obtain asymptotically exact embeddings [5]. What this means in practice (as illustrated in our numerical results for the simulated data example in the appendix) is that the manifold may be very hard to distinguish, even for a large graph. Missing data may have a similarly negative impact on discerning manifold structure. There could be substantial estimation gains in better integrating the factorisation and manifold estimation steps to overcome these difficulties.

Another limitation is that our theory restricts attention to the case of positive-definite kernels. When $\mathbf{A}$ is say a correlation or covariance matrix the positive-definite assumption on the kernel is of course natural, but when $\mathbf{A}$ is say an adjacency matrix, it is a less natural assumption. The implications of removing the positive-definite assumption are the subject of ongoing research.

## Funding Transparency Statement

Nick Whiteley and Patrick Rubin-Delanchy's research was supported by Turing Fellowships from the Alan Turing Institute. Annie Gray's research was supported by a studentship from Compass, the EPSRC Centre for Doctoral Training in Computational Statistics and Data Science.

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
