# A Theory

## A.1 Supporting results and proof of Theorem 1

**Lemma 1.** *If Assumption 1 holds, $\phi$ is injective.*

*Proof.* We prove the contrapositive. So suppose $\phi$ is not injective. Then there must exist $x, y$, with $x \neq y$, such that $\phi(x) = \phi(y)$, and hence for any $a$, $f(x, a) = \langle \phi(x), \phi(a) \rangle_2 = \langle \phi(y), \phi(a) \rangle_2 = f(y, a)$. $\square$

The inverse of $\phi$ on $\mathcal{M}$ is denoted $\phi^{-1}$. Let $\widetilde{\mathbf{H}}_{z,z'} \in \mathbb{R}^{d \times d}$ be the matrix with elements

$$\widetilde{\mathbf{H}}_{z,z'}^{ij} := \left. \frac{\partial^2 f}{\partial x^{(i)} \partial y^{(i)}} \right|_{(z,z')}.$$

**Lemma 2.** *If Assumption 2 holds, then for any $x, y \in \mathcal{Z}$, there exists $z$ on the line segment with end-points $x, y$ such that*

$$\|\phi(x) - \phi(y)\|_2^2 = \left\langle x - y, \left[ \int_0^1 \widetilde{\mathbf{H}}_{z,y+s(x-y)} \mathrm{d}s \right] (x - y) \right\rangle,$$

*where the integral is element-wise.*

*Proof.* Fix any $x, y$ in $\mathcal{Z}$. Observe from (3) and the definition of $\phi$ that for any $x, y \in \mathcal{Z}$,

$$\|\phi(x) - \phi(y)\|_2^2 = f(x, x) + f(y, y) - 2f(x, y).$$

Now define

$$g(u) := f(u, x) - f(u, y),$$

and so since $f$ is symmetric,

$$g(x) - g(y) = \|\phi(x) - \phi(y)\|_2^2.$$

By the mean value theorem, there exists $z$ on the line segment with end-points $x, y$ (i.e. $z \in \widetilde{\mathcal{Z}}$) such that

$$g(x) - g(y) = \langle \nabla g(z), x - y \rangle$$
$$= \langle \nabla_x f(z, y) - \nabla_x f(z, x), x - y \rangle$$

where $\nabla g$ is the gradient of $u \mapsto g(u)$ (with $x, y$ still considered fixed) and $\nabla_x f(z, u)$ is the gradient of $x \mapsto f(x, u)$ evaluated at $z$ (with $u$ considered fixed). Now considering the vector-valued mapping $u \mapsto \nabla_x f(z, u)$ with $z$ fixed, we have

$$\nabla_x f(z, x) - \nabla_x f(z, y) = \left[ \int_0^1 \widetilde{\mathbf{H}}_{z,y+s(x-y)} \mathrm{d}s \right] (x - y).$$

Combining the above equalities gives:

$$\|\phi(x) - \phi(y)\|_2^2 = \langle \nabla_x f(z, y) - \nabla_x f(z, x), x - y \rangle$$
$$= \left\langle x - y, \left[ \int_0^1 \widetilde{\mathbf{H}}_{z,y+s(x-y)} \mathrm{d}s \right] (x - y) \right\rangle.$$

$\square$

**Lemma 3.** *For any matrix $\mathbf{B} \in \mathbb{R}^{d \times d}$ and $z \in \mathbb{R}^d$, $|\langle z, \mathbf{B}z \rangle|^{1/2} \leq \|\mathbf{B}\|_F^{1/2} \|z\|$, where $\| \cdot \|_F$ is the Frobenius norm.*

*Proof.*

$$\langle z, \mathbf{B}z \rangle = \frac{1}{2} \langle z, (\mathbf{B} + \mathbf{B}^\top)z \rangle \leq \|z\|^2 \lambda_{\max} \leq \|z\|^2 \frac{1}{2} \|\mathbf{B} + \mathbf{B}^\top\|_F \leq \|z\|^2 \|\mathbf{B}\|_F,$$

where $\lambda_{\max}$ is the maximum eigenvalue of the symmetric matrix $(\mathbf{B} + \mathbf{B}^\top)/2$. Replacing $\mathbf{B}$ by $-\mathbf{B}$ and using $\|\mathbf{B}\|_F = \| - \mathbf{B}\|_F$ yields the lower bound $\langle z, \mathbf{B}z \rangle \geq -\|z\|^2 \|\mathbf{B}\|_F$. $\square$

**Lemma 4.** *If Assumption 2 holds, then for any $\epsilon > 0$ there exists $\delta > 0$ such that for any $x, y \in \mathcal{Z}$ such that $\|x - y\| \leq \delta$ and any $\xi, z, z'$ on the line segment with endpoints $x, y$,*

$$\left| \left\langle x - y, (\mathbf{H}_\xi - \widetilde{\mathbf{H}}_{z,z'})(x - y) \right\rangle \right| \leq \epsilon \|x - y\|^2.$$

*Proof.* For each $i, j$, since $(z, z') \mapsto \widetilde{\mathbf{H}}^{ij}_{z,z'}$ is assumed continuous on $\widetilde{\mathcal{Z}} \times \widetilde{\mathcal{Z}}$, and $\widetilde{\mathcal{Z}}$ is compact, by the Heine-Cantor theorem $(z, z') \mapsto \widetilde{\mathbf{H}}^{ij}_{z,z'}$ is in fact uniformly continuous on $\widetilde{\mathcal{Z}} \times \widetilde{\mathcal{Z}}$. Fix any $\epsilon > 0$. Using this uniform continuity, there exists $\delta > 0$ such that for any $x, y \in \mathcal{Z}$, if $\|x - y\| \leq \delta$, then for any $\xi, z, z'$ on the line-segment with end-points $x, y$,

$$\max_{i,j=1,\ldots,d} \left| \widetilde{\mathbf{H}}^{i,j}_{\xi,\xi} - \widetilde{\mathbf{H}}^{i,j}_{z,z'} \right| \leq \epsilon d^{-1},$$

and so in turn

$$\left\| \widetilde{\mathbf{H}}_{\xi,\xi} - \widetilde{\mathbf{H}}_{z,z'} \right\|_{\mathrm{F}} \leq \epsilon,$$

where $\|\cdot\|_{\mathrm{F}}$ is the Frobenius norm. Observing that $\widetilde{\mathbf{H}}_{\xi,\xi} = \mathbf{H}_z$, the result then follows from Lemma 3. $\qquad\square$

**Proposition 4.** *If Assumptions 1 and 2 hold, $\phi$ and $\phi^{-1}$ are each Lipschitz continuous with respect to the norms $\|\cdot\|$ on $\mathcal{Z}$ and $\|\cdot\|_2$ on $\mathcal{M}$.*

*Proof.* As a preliminary note that for any $z \in \mathcal{Z}$, $\mathbf{H}_z$ is symmetric and positive-definite under Assumption 2, and let $\lambda_z^{\min}, \lambda_z^{\max}$ be the minimum and maximum eigenvalues of the matrix $\mathbf{H}_z$. Since $\lambda_z^{\max} = \|\mathbf{H}_z\|_{\mathrm{sp}}$, the spectral norm of $\mathbf{H}_z$, and the reverse triangle inequality for this norm states $|\|\mathbf{H}_z\|_{\mathrm{sp}} - \|\mathbf{H}_{z'}\|_{\mathrm{sp}}| \leq \|\mathbf{H}_z - \mathbf{H}_{z'}\|_{\mathrm{sp}}$, the continuity in $z$ of the elements of $\mathbf{H}_z$ under Assumption 2 implies continuity of $z \mapsto \lambda_z^{\max}$. Similar consideration of $\lambda_z^{\min} = \|\mathbf{H}_z^{-1}\|_{\mathrm{sp}}^{-1}$ together with

$$\|\mathbf{H}_z^{-1} - \mathbf{H}_{z'}^{-1}\|_{\mathrm{sp}} \leq \|\mathbf{H}_z^{-1}\mathbf{H}_{z'} - \mathbf{I}\|_{\mathrm{sp}} \|\mathbf{H}_{z'}^{-1}\|_{\mathrm{sp}} \leq \|\mathbf{H}_z^{-1}\|_{\mathrm{sp}} \|\mathbf{H}_{z'} - \mathbf{H}_z\|_{\mathrm{sp}} \|\mathbf{H}_{z'}^{-1}\|_{\mathrm{sp}}$$

shows that $z \mapsto \lambda_z^{\min}$ is continuous. Due to the compactness of $\mathcal{Z}$, we therefore find that $\lambda^+ := \sup_{z \in \mathcal{Z}} \lambda_z^{\max} < \infty$, and $\lambda^- := \inf_{z \in \mathcal{Z}} \lambda_z^{\min} > 0$.

Our next objective in the proof of the proposition is to establish the Lipschitz continuity of $\phi$. As a first step towards this, note that it follows from the identity

$$\|\phi(x) - \phi(y)\|_2^2 = f(x, x) + f(y, y) - 2f(x, y),$$

that the continuity of $(x, y) \mapsto f(x, y)$ implies continuity in $\ell_2$ of $x \mapsto \phi(x)$. Now fix any $\epsilon_1 > 0$ and consider any $x, y \in \mathcal{Z}$. By combining Lemmas 2 and 4 there exists $\delta_1 > 0$ such that if $\|x - y\| \leq \delta_1$, there exists $z$ on the line segment with end-points $x, y$ such that:

$$\|\phi(x) - \phi(y)\|_2^2 = \left\langle x - y, \left[ \int_0^1 \widetilde{\mathbf{H}}_{z, y + s(x-y)} \mathrm{d}t \right] (x - y) \right\rangle$$

$$= \langle x - y, \mathbf{H}_z(x - y) \rangle + \int_0^1 \left\langle x - y, \left[ \widetilde{\mathbf{H}}_{z, y + s(x-y)} - \mathbf{H}_z \right] (x - y) \right\rangle \mathrm{d}s \quad (13)$$

$$\leq (\lambda^+ + \epsilon_1) \|x - y\|^2. \quad (14)$$

On the other hand if $\|x - y\| > \delta_1$,

$$\frac{\|\phi(x) - \phi(y)\|_2}{\|x - y\|} \leq c_1 \delta_1^{-1}, \quad (15)$$

where $c_1 := \sup_{x, y \in \mathcal{Z}} \|\phi(x) - \phi(y)\|_2$ is finite since $\mathcal{Z}$ is compact and $\phi$ has already been proved to be continuous in $\ell_2$. Combining (14) and (15) we obtain

$$\|\phi(x) - \phi(y)\|_2 \leq \left[ c_1 \delta_1^{-1} \vee (\lambda^+ + \epsilon_1)^{1/2} \right] \|x - y\|, \qquad \forall x, y \in \mathcal{Z}.$$

It remains to prove Lipschitz continuity of $\phi^{-1}$. Fix $\epsilon_2 \in (0, \lambda^-)$. Since $\mathcal{Z}$ is compact and $\phi$ is continuous, $\phi^{-1}$ is continuous on $\mathcal{M}$ [54, Prop. 13.26], and then also uniformly continuous by

the Heine-Cantor Theorem since $\mathcal{M}$ is compact. Putting this uniform continuity of $\phi^{-1}$ together with Lemmas 2 and 4, via the identity (13), there exists $\delta_2 > 0$ such that for any $a, b \in \mathcal{M}$, if $\|a - b\|_2 \leq \delta_2$ then
$$\|a - b\|_2^2 \geq (\lambda^- - \epsilon_2)\|\phi^{-1}(a) - \phi^{-1}(b)\|^2.$$
On the other hand, if $\|a - b\|_2 > \delta_2$,
$$\frac{\|\phi^{-1}(a) - \phi^{-1}(b)\|}{\|a - b\|_2} \leq c_2 \delta_2^{-1},$$
where $c_2 := \sup_{x,y \in \mathcal{Z}} \|x - y\|$ is finite since $\mathcal{Z}$ is compact. Therefore
$$\|\phi^{-1}(a) - \phi^{-1}(b)\| \leq \left[ c_2 \delta_2^{-1} \vee (\lambda^- - \epsilon_2)^{-1/2} \right] \|a - b\|_2, \qquad \forall a, b \in \mathcal{M}.$$
$\square$

**Lemma 5.** *If Assumptions 1 and 2 hold, then for any $a, b \in \mathcal{M}$ and a path $\gamma$ in $\mathcal{M}$ with end-points $a, b$ such that $\ell(\gamma) < \infty$, the mapping $\eta : [0, 1] \to \mathcal{Z}$ defined by $\eta_t := \phi^{-1}(\gamma_t)$ is a path in $\mathcal{Z}$ with end-points $\phi^{-1}(a), \phi^{-1}(b)$, and $l(\eta) < \infty$.*

*Proof.* By Proposition 4, $\phi^{-1}$ is continuous, which combined with the continuity of $t \mapsto \gamma_t$ implies continuity of $t \mapsto \phi^{-1}(\gamma_t)$, so $\eta$ is indeed a path in $\mathcal{Z}$, and the end points of $\eta$ are clearly $\phi^{-1}(a), \phi^{-1}(b)$. Proposition 4 establishes that moreover $\phi^{-1}$ is Lipschitz continuous, and then $l(\gamma) < \infty$ implies $l(\eta) < \infty$ due to the definition of path-length. $\square$

**Lemma 6.** *For any $a \geq 0$ and $b$ such that $|b| \leq a$,*
$$|a|^{1/2} - |b|^{1/2} \leq (a + b)^{1/2} \leq |a|^{1/2} + |b|^{1/2}.$$

*Proof.* First prove the lower bound. For any $a, b$ as in the statement, let $c = a - |b|$, so that $c \geq 0$, and set $x = |b|^{1/2}$ and $y = c^{1/2}$. Since $x, y \geq 0$, application of the Euclidean triangle inequality in $\mathbb{R}^2$ to the pair of vectors $[x \ 0]^\top, [0 \ y]^\top$ gives the fact: $(x^2 + y^2)^{1/2} \leq x + y$, hence $a^{1/2} = (|b| + c)^{1/2} \leq |b|^{1/2} + c^{1/2} = |b|^{1/2} + (a - |b|)^{1/2}$, or equivalently:
$$(a - |b|)^{1/2} \geq a^{1/2} - |b|^{1/2}. \tag{16}$$
By the reverse triangle inequality and the assumptions on $a$ and $b$,
$$(a + b)^{1/2} = |a + b|^{1/2} \geq (a - |b|)^{1/2}. \tag{17}$$
Combining (16) and (17) completes the proof of the lower bound in the statement.

For the upper-bound in the statement, let $c = a^{1/2}$ and $d = |b|^{1/2}$. Then
$$a + b \leq c^2 + d^2 = (c + d)^2 - 2cd \leq (c + d)^2,$$
which implies
$$(a + b)^{1/2} \leq a^{1/2} + |b|^{1/2}$$
as required. $\square$

*Proof of Theorem 1.* By Lemma 1, $\phi$ is injective; by Proposition 4, $\phi$ and its inverse on $\mathcal{M}$, namely $\phi^{-1}$, are Lipschitz; and by Lemma 5, for $\gamma$ and $\eta$ as in the statement, $\eta$ is a path as claimed with $l(\eta) < \infty$.

For the remainder of the proof, fix any $\epsilon > 0$. By the definition of the path-length $l(\gamma)$, there exists a partition $\widetilde{\mathcal{P}}_\epsilon$ such that
$$l(\gamma) - \epsilon/2 \leq \chi(\gamma, \widetilde{\mathcal{P}}_\epsilon) \leq l(\gamma). \tag{18}$$
Let $\mathcal{P} = (t_0, \ldots, t_n)$ be any partition, and fix any $k$ such that $1 \leq k \leq n$. By Lemma 2 there exists $z$ on the line segment with end-points $\eta_{t_{k-1}}, \eta_{t_k}$ such that
$$\|\gamma_{t_k} - \gamma_{t_{k-1}}\|_2^2 = \|\phi(\eta_{t_k}) - \phi(\eta_{t_{k-1}})\|_2^2$$
$$= \left\langle \eta_{t_k} - \eta_{t_{k-1}}, \left[ \int_0^1 \widetilde{\mathbf{H}}_{z, \eta_{t_{k-1}} + s(\eta_{t_k} - \eta_{t_{k-1}})} \mathrm{d}s \right] (\eta_{t_k} - \eta_{t_{k-1}}) \right\rangle \tag{19}$$
$$= a_k + b_k \tag{20}$$

where

$$a_k := \left\langle \eta_{t_k} - \eta_{t_{k-1}}, \mathbf{H}_{\eta_{t_{k-1}}} (\eta_{t_k} - \eta_{t_{k-1}}) \right\rangle$$

$$b_k := \int_0^1 \left\langle \eta_{t_k} - \eta_{t_{k-1}}, \left[ \widetilde{\mathbf{H}}_{z, \eta_{t_{k-1}} + s(\eta_{t_k} - \eta_{t_{k-1}})} - \mathbf{H}_{\eta_{t_{k-1}}} \right] (\eta_{t_k} - \eta_{t_{k-1}}) \right\rangle \mathrm{d}s.$$

Under Assumption 2, $\mathbf{H}_{\eta_{t_{k-1}}}$ is positive-definite, so $a_k \geq 0$, and by (19), $a_k + b_k \geq 0$, so we must have $|b_k| \leq a_k$. Lemma 6 then gives

$$a_k^{1/2} - |b_k|^{1/2} \leq (a_k + b_k)^{1/2} \leq a_k^{1/2} + |b_k|^{1/2},$$

hence:

$$\left| \|\gamma_{t_k} - \gamma_{t_{k-1}}\|_2 - a_k^{1/2} \right| \leq |b_k|^{1/2}. \tag{21}$$

By Lemma 4, there exists $\delta > 0$ such that

$$\max_{k=1,\ldots,n} \|\eta_{t_k} - \eta_{t_{k-1}}\| \leq \delta \quad \Rightarrow \quad |b_k|^{1/2} \leq \frac{\epsilon}{2} \frac{1}{l(\eta)} \|\eta_{t_k} - \eta_{t_{k-1}}\|, \quad \forall 1 \leq k \leq n. \tag{22}$$

Since $\eta$ is a path, it is continuous on the compact set $[0, 1]$, and then in fact uniformly continuous by the Heine-Cantor Theorem. Hence there exists a suitably fine partition $\mathcal{P}_{\epsilon,\delta} \supseteq \widetilde{\mathcal{P}}_\epsilon$ such that if $\mathcal{P} = (t_0, \ldots, t_n) \supseteq \mathcal{P}_{\epsilon,\delta}$, $\max_{k=1,\ldots,n} \|\eta_{t_k} - \eta_{t_{k-1}}\| \leq \delta$ and in turn from (22),

$$\sum_{k=1}^n |b_k|^{1/2} \leq \frac{\epsilon}{2} \frac{1}{l(\eta)} \sum_{k=1}^n \|\eta_{t_k} - \eta_{t_{k-1}}\| \leq \frac{\epsilon}{2}, \tag{23}$$

where the final inequality holds due to the definition of $l(\eta)$ as the length of $\eta$.

Combining (21) and (23), if again $\mathcal{P} \supseteq \mathcal{P}_{\epsilon,\delta}$,

$$\begin{aligned} \left| \chi(\gamma, \mathcal{P}) - \sum_{k=1}^n a_k^{1/2} \right| &= \left| \sum_{k=1}^n \|\gamma_{t_k} - \gamma_{t_{k-1}}\|_2 - a_k^{1/2} \right| \\ &\leq \sum_{k=1}^n |b_k|^{1/2} \\ &\leq \frac{\epsilon}{2}. \end{aligned} \tag{24}$$

Recalling from (18) the defining property of $\widetilde{\mathcal{P}}_\epsilon$ and using $\mathcal{P} \supseteq \mathcal{P}_{\epsilon,\delta} \supseteq \widetilde{\mathcal{P}}_\epsilon$, the triangle inequality for the $\|\cdot\|_2$ norm gives

$$l(\gamma) - \frac{\epsilon}{2} \leq \chi(\gamma, \widetilde{\mathcal{P}}_\epsilon) \leq \chi(\gamma, \mathcal{P}) \leq l(\gamma).$$

Combined with (24), we finally obtain that if $\mathcal{P} \supseteq \mathcal{P}_{\epsilon,\delta}$,

$$l(\gamma) - \epsilon \leq \sum_{k=1}^n a_k^{1/2} \leq l(\gamma) + \frac{\epsilon}{2}$$

and the proof of (4) is completed by taking $\mathcal{P}_\epsilon$ as appears in the statement to be $\mathcal{P}_{\epsilon,\delta}$.

The first equality in (5) can be proved by a standard argument - e.g., [47, p.137]. The second inequality in (5) is proved by passing to the limit of the summation in (4) along any sequence of partitions $\mathcal{P}^{(m)} = (t_0^{(m)}, \ldots, t_{n(m)}^{(m)})$, $m \geq 1$, with $\mathcal{P}^{(m)} \supseteq \mathcal{P}_{\epsilon,\delta}$ such that $\lim_{m \to \infty} \max_{k=1,\ldots,n(m)} |t_k^{(m)} - t_{k-1}^{(m)}| = 0$. $\qquad \square$

## A.2 Deriving geodesic distances from (4) rather than from (5)

Recall from Section 3.3 that the general strategy to derive the geodesic distance associated with each family of kernels (translation invariant, inner-product, additive) is:

(i) identify a lower bound on $l(\gamma)$ which holds over all paths $\gamma$ in $\mathcal{M}$ which have generic end-points $a, b \in \mathcal{M}$ in common, then

(ii) show there exists a path whose length is equal to this lower bound.

In Section 3.3 this strategy was executed for each family of kernels starting from the expression for $l(\gamma)$ given in (5). In the proofs of Lemmas 7–9 below we show how step (i) is performed if we start not from (5) but rather from (4), the latter being more general because continuous differentiability of the paths is relaxed to continuity. The key message of these three lemmas regarding step (i) is that we obtain exactly the same lower bounds on $l(\gamma)$ as are derived from (5) in Section 3.3. The reader is directed to Section 3.3 for the details of how Assumption 2 is verified for each family of kernels; to avoid repetition we don't re-state all those details here.

**Lemma 7.** *Consider the family of translation invariant kernels described in Section 3.3 and let $\mathbf{G}$ be as defined there. For any $a, b \in \mathcal{M}$ and any path $\gamma \in \mathcal{M}$ with end-points $a, b$,*

$$l(\gamma) \geq \|\mathbf{G}^\top[\phi^{-1}(b) - \phi^{-1}(a)]\|.$$

*If we define $\tilde{\eta}$ to be the path in $\mathcal{Z}$ given by*

$$\tilde{\eta}_t := \phi^{-1}(a) + t[\phi^{-1}(b) - \phi^{-1}(a)], \quad t \in [0, 1],$$

*then $\tilde{\gamma}$ defined by $\tilde{\gamma}_t := \phi(\tilde{\eta}_t)$ is a path in $\mathcal{M}$ with end-points $a, b$ and $l(\tilde{\gamma}) = \|\mathbf{G}^\top[\phi^{-1}(b) - \phi^{-1}(a)]\|$.*

*Proof.* Applying Theorem 1, fix any $\epsilon > 0$ and let $\mathcal{P}_\epsilon$ be a partition such that for any partition $\mathcal{P} = (t_0, \ldots, t_n)$ satisfying $\mathcal{P}_\epsilon \subseteq \mathcal{P}$,

$$\left| l(\gamma) - \sum_{k=1}^{n} \left\langle \eta_{t_k} - \eta_{t_{k-1}}, \mathbf{H}_{\eta_{t_{k-1}}}(\eta_{t_k} - \eta_{t_{k-1}}) \right\rangle^{1/2} \right| \leq \epsilon. \tag{25}$$

Recalling from Section 3.3 that for this family of translation invariant kernels $\mathbf{H}_z = \mathbf{G}\mathbf{G}^\top$ for all $z \in \mathcal{Z}$, the triangle inequality for the $\|\cdot\|$ norm combined with (25) gives

$$l(\gamma) \geq -\epsilon + \sum_{k=1}^{n} \left\langle \eta_{t_k} - \eta_{t_{k-1}}, \mathbf{H}_{\eta_{t_{k-1}}}(\eta_{t_k} - \eta_{t_{k-1}}) \right\rangle^{1/2}.$$

$$= -\epsilon + \sum_{k=1}^{n} \|\mathbf{G}^\top(\eta_{t_k} - \eta_{t_{k-1}})\|$$

$$\geq -\epsilon + \left\| \sum_{k=1}^{n} \mathbf{G}^\top(\eta_{t_k} - \eta_{t_{k-1}}) \right\|$$

$$= -\epsilon + \|\mathbf{G}^\top(\eta_1 - \eta_0)\|$$

$$= -\epsilon + \|\mathbf{G}^\top[\phi^{-1}(b) - \phi^{-1}(a)]\|.$$

The proof of the lower bound in the statement is then complete since $\epsilon$ was arbitrary. To complete the proof of the lemma, observe that from the definition of $\tilde{\eta}$ in the statement,

$$\sum_{k=1}^{n} \left\langle \tilde{\eta}_{t_k} - \tilde{\eta}_{t_{k-1}}, \mathbf{H}_{\tilde{\eta}_{t_{k-1}}}(\tilde{\eta}_{t_k} - \tilde{\eta}_{t_{k-1}}) \right\rangle^{1/2}$$

$$= \sum_{k=1}^{n} \|(t_k - t_{k-1})\mathbf{G}^\top[\phi^{-1}(b) - \phi^{-1}(a)]\|$$

$$= \|\mathbf{G}^\top[\phi^{-1}(b) - \phi^{-1}(a)]\| \sum_{k=1}^{n} (t_k - t_{k-1})$$

$$= \|\mathbf{G}^\top[\phi^{-1}(b) - \phi^{-1}(a)]\|,$$

and the proof of the lemma is then complete, because $\epsilon$ in (25) being arbitrary implies $l(\tilde{\gamma}) = \|\mathbf{G}^\top[\phi^{-1}(b) - \phi^{-1}(a)]\|$.

$\square$

**Lemma 8.** *Consider the family of inner-product kernels of the form $f(x,y) = g(\langle x, y \rangle)$ as described in Section 3.3 where $g'(1) > 0$. For any $a, b \in \mathcal{M}$ and any path $\gamma \in \mathcal{M}$ with end-points $a, b$,*

$$l(\gamma) \geq g'(1)^{1/2} \arccos \left\langle \phi^{-1}(a), \phi^{-1}(b) \right\rangle.$$

*If $\tilde{\eta}$ is a shortest circular arc in $\mathcal{Z}$ with end-points $\phi^{-1}(a), \phi^{-1}(b)$, then $\tilde{\gamma}$ defined by $\tilde{\gamma}_t := \phi(\tilde{\eta}_t)$ satisfies $l(\tilde{\gamma}) = g'(1)^{1/2} \arccos \left\langle \phi^{-1}(a), \phi^{-1}(b) \right\rangle$.*

*Proof.* As usual, let $\eta$ be the path in $\mathcal{Z}$ defined by $\eta_t := \phi^{-1}(\gamma_t)$. Then from the definition of path-length and the triangle inequality for the $\| \cdot \|$ norm, for any $\delta > 0$, there exists a partition $\mathcal{P}_\delta^\star$ such that for any $\mathcal{P}^\star = (t_0^\star, \ldots, t_n^\star)$ satisfying $\mathcal{P}_\delta^\star \subseteq \mathcal{P}^\star$,

$$\sum_{k=1}^n \|\eta_{t_k^\star} - \eta_{t_{k-1}^\star}\| \geq l(\eta) - \frac{\delta}{g'(1)^{1/2}}. \tag{26}$$

Fix any $\epsilon > 0$ and let $\mathcal{P}_\epsilon$ be as in Theorem 1 and then take $\mathcal{P} = (t_0, \ldots, t_n)$ to be defined by $\mathcal{P} = \mathcal{P}_\epsilon \cup \mathcal{P}_\delta^\star$, so by construction we have simultaneously $\mathcal{P}_\epsilon \subseteq \mathcal{P}$ and $\mathcal{P}_\delta^\star \subseteq \mathcal{P}$. Then from Theorem 1,

$$\left| l(\gamma) - \sum_{k=1}^n \left\langle \eta_{t_k} - \eta_{t_{k-1}}, \mathbf{H}_{\eta_{t_{k-1}}} (\eta_{t_k} - \eta_{t_{k-1}}) \right\rangle^{1/2} \right| \leq \epsilon. \tag{27}$$

Combined with the fact that for this family of kernels $\mathbf{H}_z = g'(1)\mathbf{I} + g''(1) z z^\top$ where $g'(1) > 0$ and $g''(1) \geq 0$, we obtain

$$
\begin{aligned}
l(\gamma) &\geq -\epsilon + \sum_{k=1}^n \left\langle \eta_{t_k} - \eta_{t_{k-1}}, \mathbf{H}_{\eta_{t_{k-1}}} (\eta_{t_k} - \eta_{t_{k-1}}) \right\rangle^{1/2} \\
&= -\epsilon + \sum_{k=1}^n \left( g'(1)\|\eta_{t_k} - \eta_{t_{k-1}}\|^2 + g''(1)|\langle \eta_{t_k} - \eta_{t_{k-1}}, z \rangle|^2 \right)^{1/2} \\
&\geq -\epsilon + g'(1)^{1/2} \sum_{k=1}^n \|\eta_{t_k} - \eta_{t_{k-1}}\| \\
&\geq -\epsilon + g'(1)^{1/2} l(\eta) - \delta,
\end{aligned}
$$

where the penultimate inequality uses $g''(1) \geq 0$ and the final inequality holds by taking $\mathcal{P}^\star$ in (26) to be $\mathcal{P}$. Since $\epsilon$ and $\delta$ were arbitrary, we have shown $l(\gamma) \geq g'(1)^{1/2} l(\eta)$. Recall that here $\eta$ is a path in $\mathcal{Z} = \{\|x\| \in \mathbb{R}^d : \|x\| = 1\}$ with end-points $\phi^{-1}(a), \phi^{-1}(b)$. Hence $l(\eta)$ is lower-bounded by the Euclidean geodesic distance in $\mathcal{Z}$ between $\phi^{-1}(a)$ and $\phi^{-1}(b)$, which is $\arccos \left\langle \phi^{-1}(a), \phi^{-1}(b) \right\rangle$ because $\mathcal{Z}$ is a radius-1 sphere centered at the origin.

With $\tilde{\eta}$ and $\tilde{\gamma}$ as defined in the statement, taking $\eta$ in (27) to be $\tilde{\eta}$, refining $\mathcal{P}$ and using $\langle \tilde{\eta}_t, \dot{\tilde{\eta}}_t \rangle = 0$ (see discussion in Section 3.3) we find $l(\tilde{\gamma}) = g'(1)^{1/2} l(\tilde{\eta})$, where by definition of $\tilde{\eta}$, $l(\tilde{\eta}) = \arccos \left\langle \phi^{-1}(a), \phi^{-1}(b) \right\rangle$.

$\square$

**Lemma 9.** *Consider the family of additive kernels described in Section 3.3. For any $a, b \in \mathcal{M}$ and any path $\gamma \in \mathcal{M}$ with end-points $a, b$,*

$$l(\gamma) \geq \|\psi \circ \phi^{-1}(b) - \psi \circ \phi^{-1}(a)\|.$$

*If we define $\tilde{\zeta}_t := \psi \circ \phi^{-1}(a) + t[\psi \circ \phi^{-1}(b) - \psi \circ \phi^{-1}(a)]$ and let $\tilde{\gamma}$ be defined by $\tilde{\gamma}_t := \phi \circ \psi^{-1}(\zeta_t)$, then $l(\tilde{\gamma}) = \|\psi \circ \phi^{-1}(b) - \psi \circ \phi^{-1}(a)\|$.*

*Proof.* The compactness of $\mathcal{Z}$ and the continuity of $z \mapsto \left. \frac{\partial^2 f_i}{\partial x^{(i)} y^{(i)}} \right|_{(z,z)}^{1/2}$ for each $i = 1, \ldots, d$ implies the uniform-continuity of the latter by the Heine-Cantor theorem. Hence for any $\delta_1 > 0$, there exists $\delta_2 > 0$ such that for all $i = 1, \ldots, d$ and $z_1^{(i)}, z_2^{(i)} \in \mathcal{Z}_i$,

$$|z_1^{(i)} - z_2^{(i)}| \leq \delta_2 \quad \Rightarrow \quad \alpha_i^{1/2} \left| \left. \frac{\partial^2 f_i}{\partial x^{(i)} y^{(i)}} \right|_{(z_1^{(i)}, z_1^{(i)})}^{1/2} - \left. \frac{\partial^2 f_i}{\partial x^{(i)} y^{(i)}} \right|_{(z_2^{(i)}, z_2^{(i)})}^{1/2} \right| \leq \frac{\delta_1}{l(\eta)}. \tag{28}$$

Fix any $\epsilon > 0$ and let $\mathcal{P}_\epsilon$ be as in Theorem 1. We now claim there exists a partition $\mathcal{P} = (t_0, \ldots, t_n)$ satisfying simultaneously $\mathcal{P}_\epsilon \subseteq \mathcal{P}$ and

$$\max_{i=1,\ldots,d} \max_{k=1,\ldots,n} |\eta_{t_k}^{(i)} - \eta_{t_{k-1}}^{(i)}| \leq \delta_2. \tag{29}$$

To see that such a partition exists, note that with $\eta_t := \phi^{-1}(\gamma_t)$, $t \mapsto \eta_t$ is continuous on the compact set $[0,1]$ hence uniformly continuous by the Heine-Cantor theorem. Thus for any $s, t \in [0,1]$ sufficiently close to each other, $\|\eta_s - \eta_t\|$ can be made less than or equal to $\delta_2$, which implies $\max_{i=1,\ldots,d} |\eta_s^{(i)} - \eta_t^{(i)}| \leq \delta_2$. Thus starting from $\mathcal{P}_\epsilon$, if we subsequently add points to this partition until $\max_{k=1,\ldots,n} |t_k - t_{k-1}|$ is sufficiently small then we will arrive at a partition $\mathcal{P}$ with the required properties, as claimed.

Now with this partition $\mathcal{P}$ in hand, fix any $i = 1, \ldots, d$ and $k = 1, \ldots, n$. We then have

$$\zeta_{t_k}^{(i)} - \zeta_{t_{k-1}}^{(i)} = \psi_i(\eta_{t_k}^{(i)}) - \psi_i(\eta_{t_{k-1}}^{(i)}) \tag{30}$$

$$= \alpha_i^{1/2} \int_{\eta_{t_{k-1}}^{(i)}}^{\eta_{t_k}^{(i)}} \left. \frac{\partial^2 f_i}{\partial x^{(i)} y^{(i)}} \right|_{(\xi, \xi)}^{1/2} \, d\xi$$

$$= \alpha_i^{1/2} \int_{\eta_{t_{k-1}}^{(i)}}^{\eta_{t_k}^{(i)}} \left[ \left. \frac{\partial^2 f_i}{\partial x^{(i)} y^{(i)}} \right|_{(\xi, \xi)}^{1/2} - \left. \frac{\partial^2 f_i}{\partial x^{(i)} y^{(i)}} \right|_{(\eta_{t_{k-1}}^{(i)}, \eta_{t_{k-1}}^{(i)})}^{1/2} \right] \, d\xi$$

$$+ \alpha_i^{1/2} \left. \frac{\partial^2 f_i}{\partial x^{(i)} y^{(i)}} \right|_{(\eta_{t_{k-1}}^{(i)}, \eta_{t_{k-1}}^{(i)})}^{1/2} (\eta_{t_k}^{(i)} - \eta_{t_{k-1}}^{(i)})$$

$$\leq \left[ \frac{\delta_1}{l(\eta)} + \alpha_i^{1/2} \left. \frac{\partial^2 f_i}{\partial x^{(i)} y^{(i)}} \right|_{(\eta_{t_{k-1}}^{(i)}, \eta_{t_{k-1}}^{(i)})}^{1/2} \right] (\eta_{t_k}^{(i)} - \eta_{t_{k-1}}^{(i)}), \tag{31}$$

where the upper bound is due to (28)-(29). Squaring both sides, summing over $i$ and then applying the triangle inequality gives

$$\|\mathbf{H}_{\eta_{t_{k-1}}}^{(1/2)} (\eta_{t_k} - \eta_{t_{k-1}})\| \geq \|\zeta_{t_k} - \zeta_{t_{k-1}}\| - \frac{\delta_1}{l(\eta)} \|\eta_{t_k} - \eta_{t_{k-1}}\|,$$

where $\mathbf{H}_{\eta_{t_{k-1}}}^{(1/2)}$ is the diagonal matrix with $i$th diagonal element equal to $\alpha_i^{1/2} \left. \frac{\partial^2 f_i}{\partial x^{(i)} y^{(i)}} \right|_{(\eta_{t_{k-1}}^{(i)}, \eta_{t_{k-1}}^{(i)})}^{1/2}$.

Summing over $k = 1, \ldots, n$ and using the definition of $l(\eta)$ gives

$$\sum_{k=1}^n \|\mathbf{H}_{\eta_{t_{k-1}}}^{(1/2)} (\eta_{t_k} - \eta_{t_{k-1}})\| \geq -\delta_1 + \sum_{k=1}^n \|\zeta_{t_k} - \zeta_{t_{k-1}}\|.$$

Combined with the relationship (4) from Theorem 1 and yet another application of the triangle inequality we thus find

$$l(\gamma) \geq -\epsilon - \delta_1 + \sum_{k=1}^n \|\zeta_{t_k} - \zeta_{t_{k-1}}\|$$

$$\geq -\epsilon - \delta_1 + \|\zeta_1 - \zeta_0\|$$

$$= -\epsilon - \delta_1 + \|\psi \circ \phi^{-1}(b) - \psi \circ \phi^{-1}(a)\|.$$

The proof of the lower bound in the statement is complete since $\epsilon$ and $\delta_1$ are arbitrary. In order to complete the proof of the lemma, observe that (28) combined with the same decomposition in (30)-(31) yields an accompanying lower bound on $\zeta_{t_k}^{(i)} - \zeta_{t_{k-1}}^{(i)}$, from which it follows that

$$\left| \sum_{k=1}^n \|\mathbf{H}_{\eta_{t_{k-1}}}^{(1/2)} (\eta_{t_k} - \eta_{t_{k-1}})\| - \sum_{k=1}^n \|\zeta_{t_k} - \zeta_{t_{k-1}}\| \right| \leq \delta_1.$$

Substituting $\tilde{\zeta}$ as defined in the statement of the lemma in place of $\zeta$, and replacing $\eta$ by $\tilde{\eta}$ defined by $\tilde{\eta}_t := \psi^{-1}(\tilde{\zeta}_t)$, then using the fact that $\delta_1$ is arbitrary we find via (4) in Theorem 1 that $l(\tilde{\gamma}) = \|\psi \circ \phi^{-1}(b) - \psi \circ \phi^{-1}(a)\|$.

$\square$

# B   Supplementary experiments

Codes for the experiments reported in the main part of the paper and those in this appendix are available at https://github.com/anniegray52/graphs.

The R packages used in this paper are (with license details therein, see github repository for code): data.table, RSpectra, igraph, plotly, Matrix, MASS, irlba, ggplot2, ggrepel, umap, Rtsne, lpSolve, spatstat, ggsci, cccd, R.utils, tidyverse, gridExtra, rgl, plot3D. The Python modules used in this paper are (with license details therein, see github repository for code): networkx, pandas, nodevectors, random.

Data on the airports (e.g. their continent) was downloaded from https://ourairports.com/data/.

## B.1   Supplementary figures for simulated data example

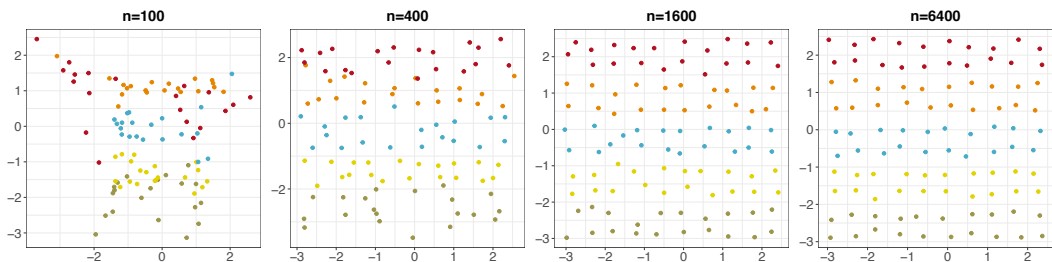

Figure 5: Latent position recovery in the sparse regime by spectral embedding followed by Isomap for increasing $n$ and increasing sparsity. To aid visualisation, all plots display a subset of 100 estimated positions corresponding to true positions on a sub-grid which is common across $n$. Estimated positions are coloured according to their true $y$-coordinate.

## B.2   Supplementary figures and discussion for flight network example

A gap appears to form between North and South America which, as a first hypothesis, we put down to the well-publicised suspension of all immigration into the US in April 2020. To measure this gap we use the Earth Mover's distance between the point clouds belonging to the two continents (with thanks to Dr. Louis Gammelgaard Jensen for the code), *using the approximate geodesic distances* of the $\epsilon$-neighbourhood graph (i.e., before dimension reduction). While this distance does explode in April 2020, as shown in Figure 12 (Appendix), we found the proposed explanation to be incomplete, because Australia and New Zealand imposed similar measures at the time, whereas the distance between Oceania and the rest of the world, computed in the same way, does not explode. Revisiting the facts [2], while the countries mentioned above closed their borders to nonresidents, the continents of South America and Africa arguably imposed more severe measures, with large numbers of countries fully suspending flights. This explanation seems more likely, as on re-inspection we find a large jump in Earth mover's distance, over April, between *both* of those continents and the rest of the world, as shown in Figure 13.

## B.3   Supplementary figures and discussion for temperature correlation example

Further to the discussion in Section 4, Figure 14 illustrates two important findings in the case $p = 2$ and $d = 1$, under the setup for this temperature correlation example described in section 4: firstly that there is no clear relationship between longitude and position along the manifold, and secondly a non-monotone relationship between longitude and estimated latent position. Both these observations are in marked contrast with the results in Figure 4 for latitude.

We then consider the case $p = 3$ and $d = 2$. Figure 15 shows the spectral embedding. Again the point-cloud is concentrated around a curved manifold. In Figure 16 we plot latitude and longitude against each of the $d = 2$ coordinates of the estimated latent positions. As in the $p = 2, d = 1$ case,

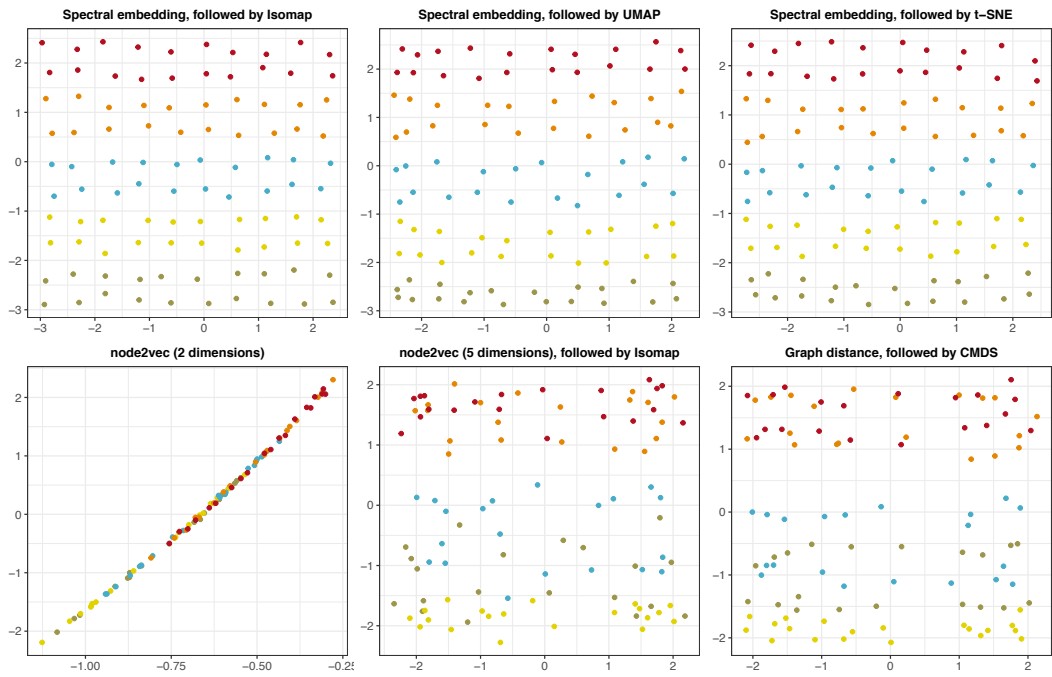

Figure 6: Latent position recovery in a sparse regime using different combinations of techniques with $n = 6400$. The first row contains spectral embedding followed by different nonlinear dimension reduction techniques and the second row contains node2vec, node2vec followed by Isomap, and we have attempted recovery using graph distances [11]. To aid visualisation, all plots will display a subset of 100 on a fixed sub-grid, coloured according to their true location.

we find a clear monotone relationship between latitude and the first component of estimated position but no such relationship between longitude and the second component.

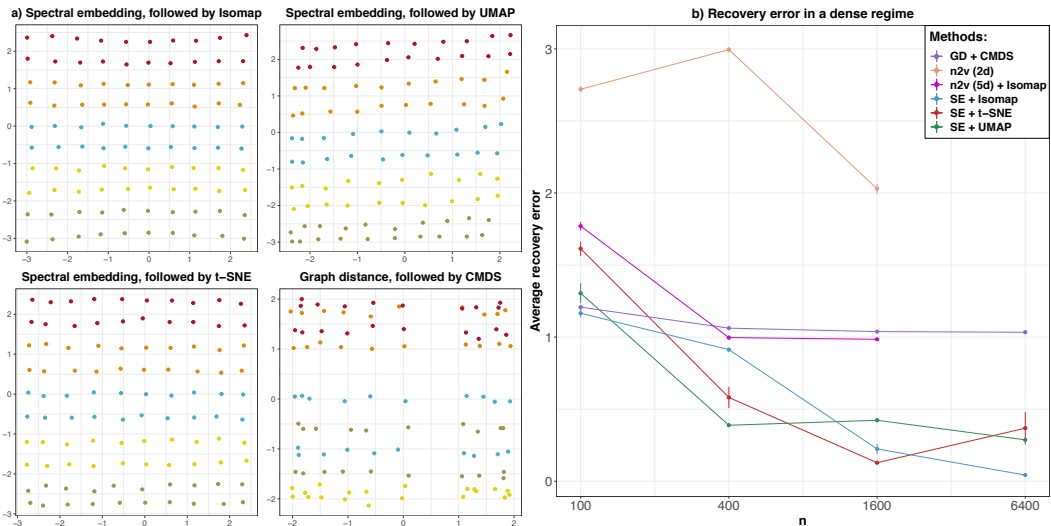

Figure 7: Latent position recovery in a dense regime. a) Recovery using different combinations of technique with $n = 6400$. b) Average recovery error. The recovery error is an average over nodes and over 100 simulations, with two standard errors shown as vertical bars. Computational issues precluded showing node2vec for $n = 6400$. (SE = spectral embedding, GD = graph distance and n2v = node2vec.)

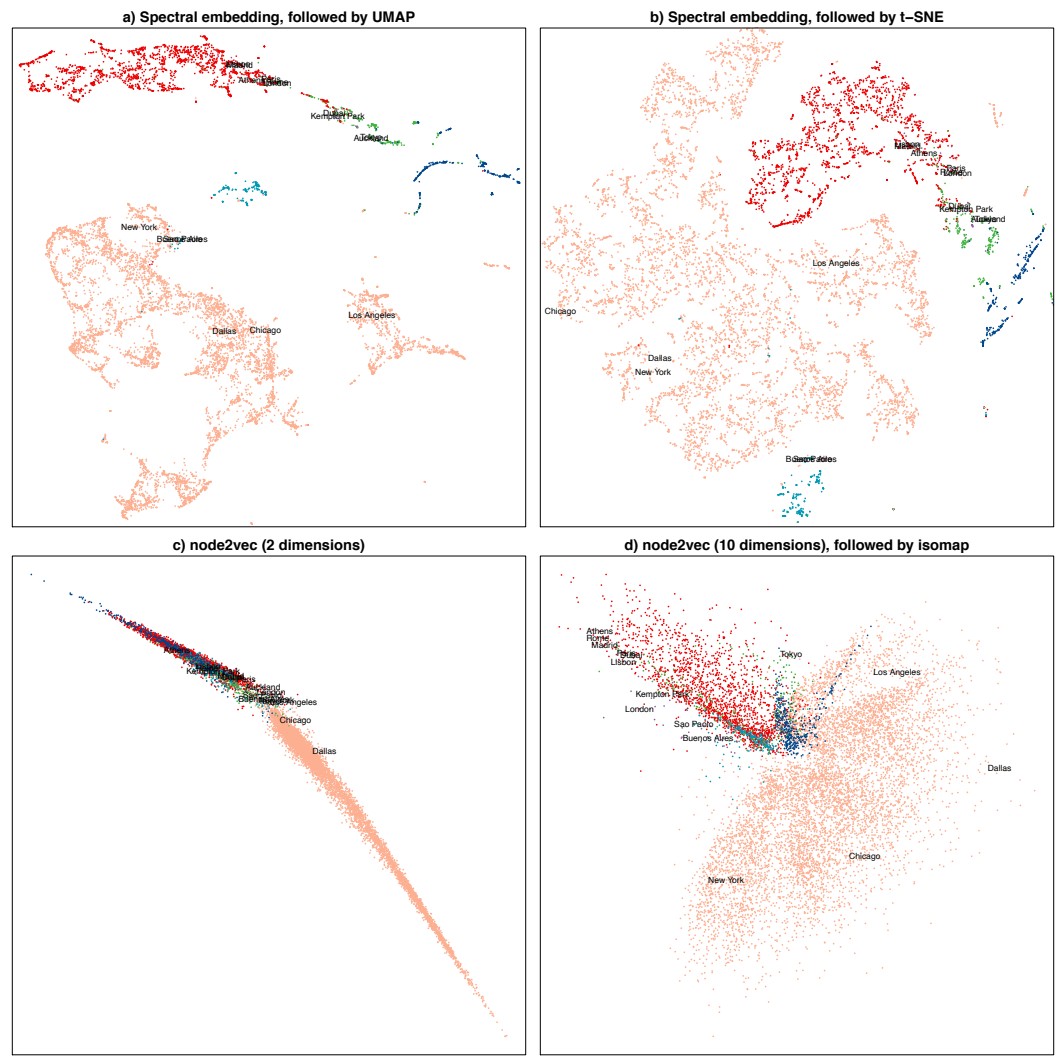

Figure 8: Visualisation of the global flight network over January 2020 by alternative combinations of techniques. The colours indicate continents (NA = North America, EU = Europe, AS = Asia, AF = Africa, OC = Oceania) and a spread of cities with high-traffic airports are labelled (to reduce clutter, only a selection of the cities in Figure 3 are shown).

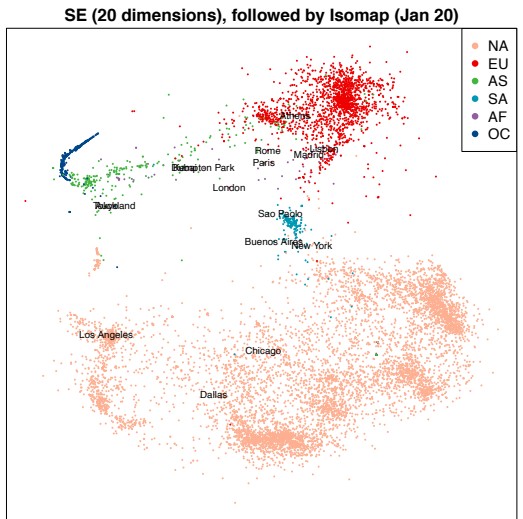

Figure 9: Visualisation of the global flight network. Spectral embedding into 20 dimensions followed by Isomap. The colours indicate continents (NA = North America, EU = Europe, AS = Asia, AF = Africa, OC = Oceania) and a spread of cities with high-traffic airports are labelled.

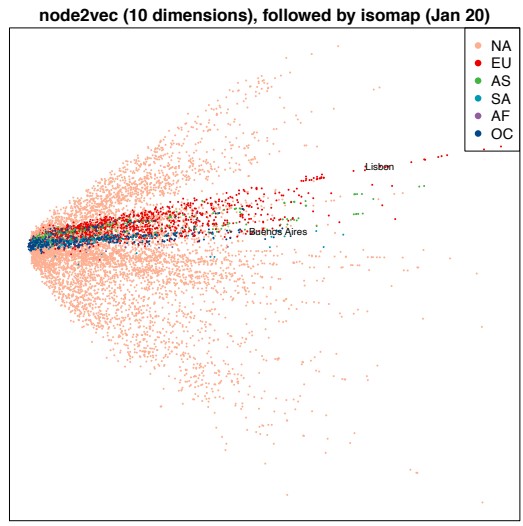

Figure 10: Visualisation of the global flight network. Node2vec into 20 dimensions followed by Isomap. The colours indicate continents (NA = North America, EU = Europe, AS = Asia, AF = Africa, OC = Oceania) and a spread of cities with high-traffic airports are labelled.

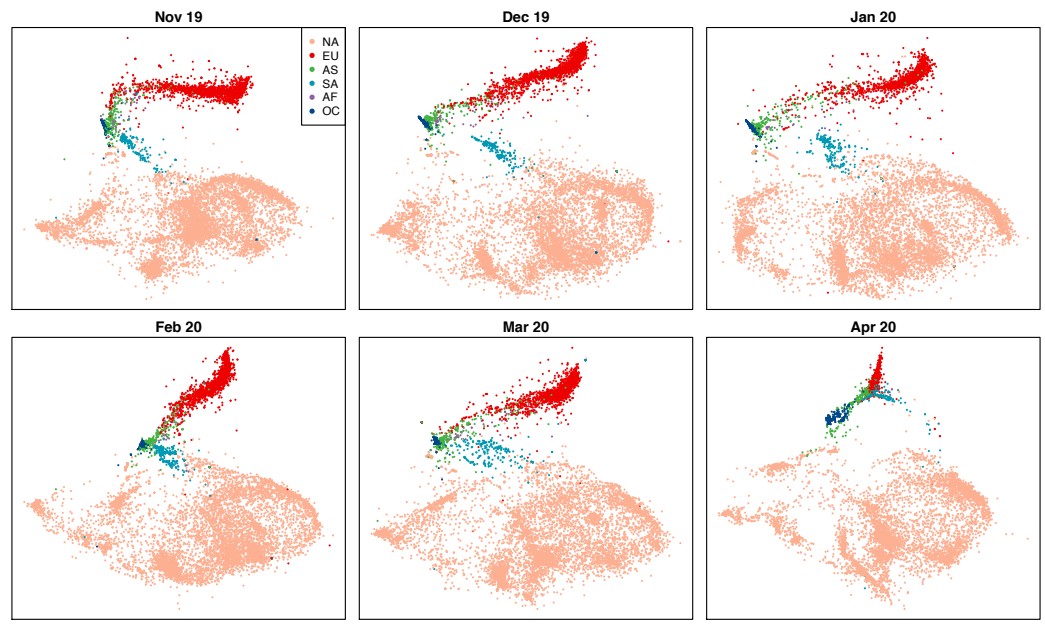

Figure 11: Visualisation of the global flight network over time: nonlinear dimension reduction of each spectral embedding using Isomap. The colours indicate continents (NA = North America, EU = Europe, AS = Asia, AF = Africa, OC = Oceania) and a spread of cities with high-traffic airports are labelled. An important structural change is observed in April 2020.

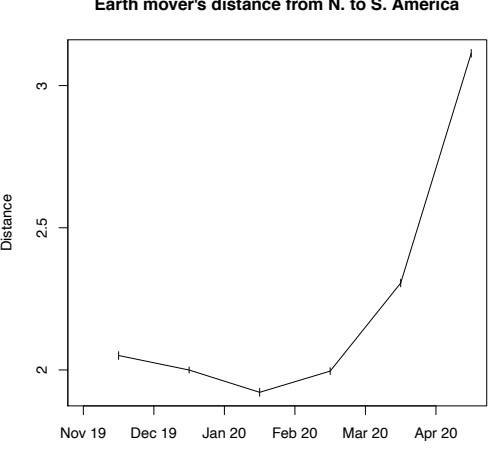

Figure 12: Earth mover's distance between North and South America, as inferred from the approximate geodesic distances given by the $\epsilon$-neighbourhood graph. In each of 100 Monte Carlo iterations, the Earth Mover's distance is computed based on 100 points randomly selected from each continent. We plot the average, with 2 standard errors in either direction indicated by the vertical bars.

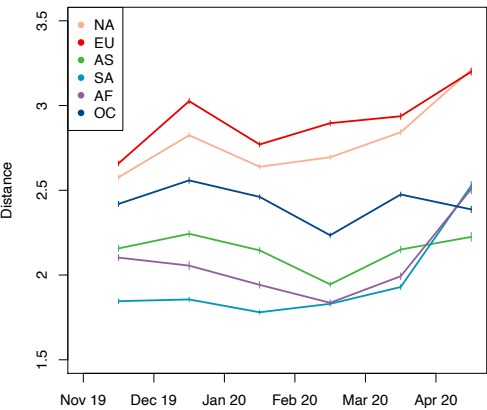

Figure 13: Earth mover's distance from each continent to the rest of the world, as inferred from the approximate geodesic distances given by the $\epsilon$-neighbourhood graph. In each of 100 Monte Carlo iterations, the Earth mover's distance is computed based on 100 points (airports) randomly selected from the continent of interest, and 100 points (airports) from the rest of the world. We plot the average, for each continent, with 2 standard errors in either direction indicated by the vertical bars.

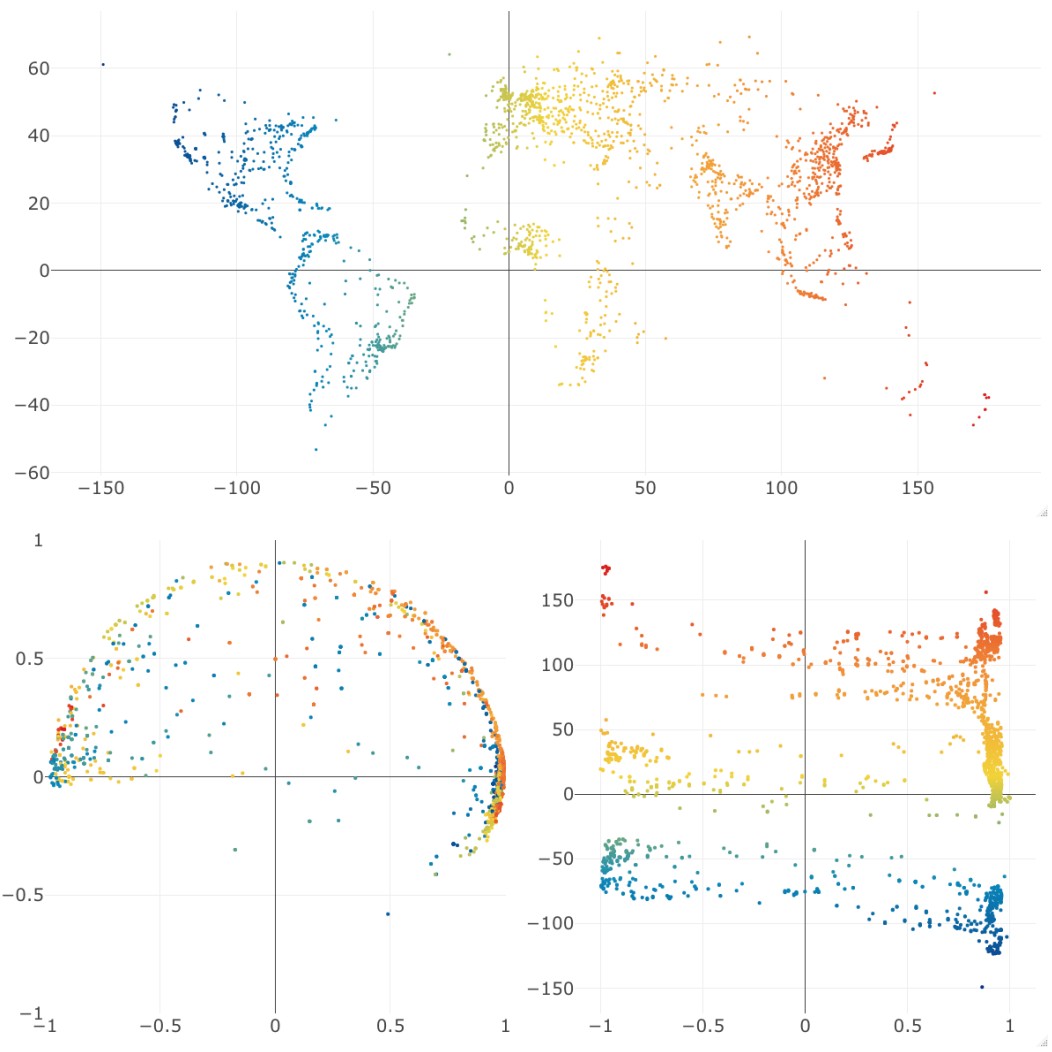

Figure 14: Temperature correlation example. Top: city locations. Bottom left: spectral embedding with $p = 2$. Bottom right: true longitude (vertical axis) vs. estimated latent position (horizontal axis) with $d = 1$. In all plots, points are coloured by longitudes of the corresponding cities.

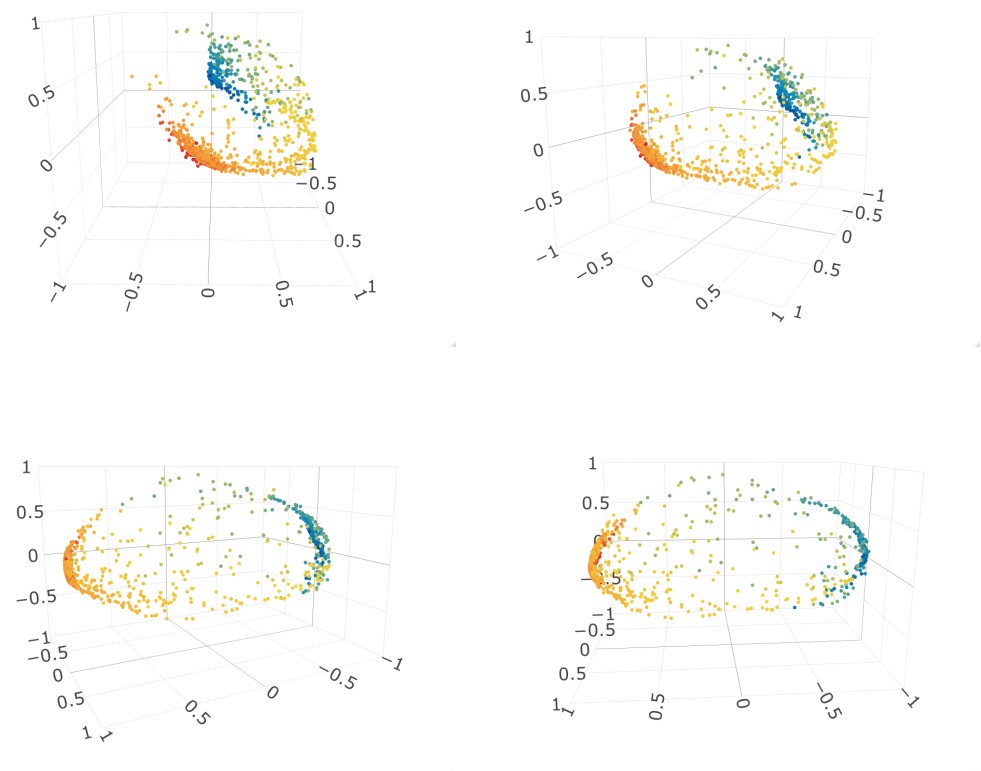

Figure 15: Temperature correlation example. Four views of the spectral embedding with $p = 3$. In all plots, points are coloured by latitudes of the corresponding cities.

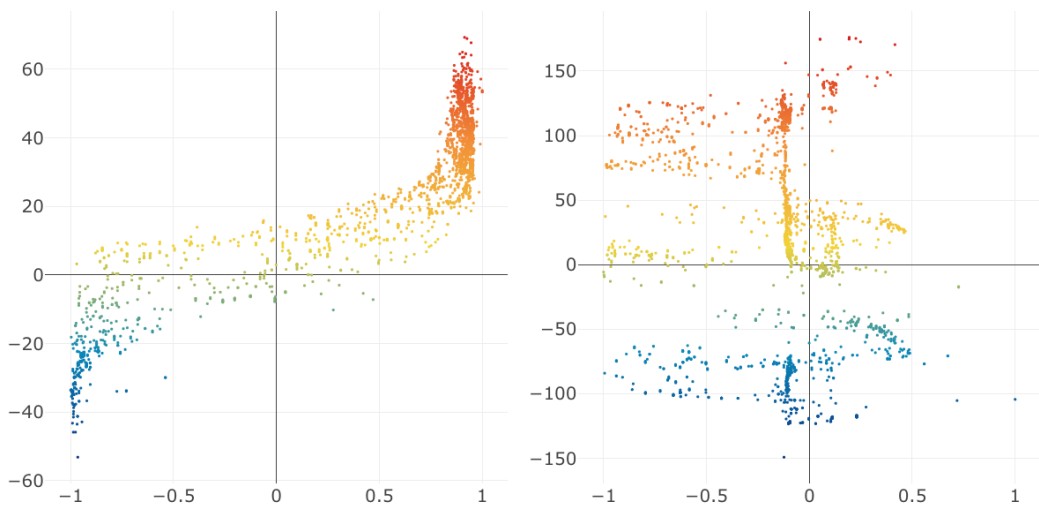

Figure 16: Temperature correlation example. Results for spectral embedding with $p = 3$ followed by Isomap with $d = 2$. Left: latitude (vertical axis) vs. estimated latent coordinate $\hat{Z}_i^{(1)}$ (horizontal axis) coloured by latitude. Right: longitude (vertical axis) vs. estimated latent coordinate $\hat{Z}_i^{(2)}$ (horizontal axis) coloured by longitude.