# OpenReview forum: "Matrix factorisation and the interpretation of geodesic distance"
_NeurIPS.cc/2021/Conference — NeurIPS 2021 Poster_

### Official Review · Reviewer_2QqF · 2021-07-10

**Rating:** 6
**Confidence:** 2

**Summary:**

The paper considers the problem of recovering a notion of distance between the nodes of a graph. The authors write their main problem very abstractly and provide an excellent description of particular A's, f's, and Z_i's that are of interest in a wide range of applications. The main contribution is to prove that a low-dimensional embedding is related to a topological manifold.  In this regard, it is a follow-up to results by Rubin-Delanchy in [21] from 2020.  The paper seems to be well written, though I was not able to follow all the technical details in the supplementary material as it is pretty far from my expertise.

**Limitations And Societal Impact:**

While the authors did make assumptions and the methodology does have limitations, the authors did not address the consequences of these limitations in terms of the potential applications.

**Main Review:**

The authors start with problem statement (1) but later make a few more assumptions on f (e.g., pos. def., f is C^2, Hessian is pos. def., and a non-degeneracy condition). I think the authors will want to notify the readers at least that those assumptions are coming.  The authors mentioned that it is technically challenging to extend their results to the case when f is symmetric but not positive definite. What are the issues there? One issue is that the eigenvalue decomposition (a.k.a. Mercer expansion for pos. def. functions) is no longer uniformly and absolutely convergent to f.  I am under the impression that if one assumes that f has a little more regularity (e.g., Lipschitz continuous in both variables separately), one can keep the uniform and absolutely convergent properties for symmetric f. I am not aware of a particular reference, though, but Schmidt, Weyl, Smithies, Hammerstein, and Brown were interested in this.

Theorem 1 is an excellent result that tells us that finding the geodesic distance in M is equivalent to minimizing path-lengths in Z under the Riemannian metric.  Section 3.3 is welcomed as it shows the potential use of that connection. Can the authors add a comment on why f needs to be C^2?

The practical applications also look impressive as a non-expert in the field of spectral embedding.

**Time Spent Reviewing:**

3

---

> ### Author Response · Authors · 2021-08-10
> **Response to reviewer 2QqF.**
>
> * The authors start with problem statement (1) but later make a few more assumptions on f (e.g., pos. def., f is C^2, Hessian is pos. def., and a non-degeneracy condition). I think the authors will want to notify the readers at least that those assumptions are coming.
>
> We will do this in the revision.
>
> * The authors mentioned that it is technically challenging to extend their results to the case when f is symmetric but not positive definite. What are the issues there?
>
> As the reviewer goes on to mention, indeed a Mercer-like expansion of the kernel holds under appropriate smoothness assumptions, but there are obstacles downstream: when relaxing the positive-definite assumption on the kernel, one needs to alter the definition of the $\phi$ to $\phi(z):=[|\lambda_1|^{1/2}u_1(z)\;|\lambda_2|^{1/2}u_2(z)\;\cdots]^T$, i.e. take the absolute values of the eigenvalues. Loosely speaking, this is needed for a result like eq. (2) to hold. However one then loses the fundamental property that in the positive-definite case $f(z,z^\prime)=\langle\phi(z),\phi(z^{\prime})\rangle_2$, which invalidates many of the arguments made in the proof of Theorem 1. We do not think this is insurmountable and we're looking into it.
>
> * Can the authors add a comment on why f needs to be C^2?.
>
> The continuity of the second derivatives implies that $\frac{\partial^2 f}{  \partial x\partial y}=\frac{\partial^2 f}{ \partial y \partial x}$ (commonly known as Young's theorem). This implies that the matrix $\mathbf{H}_z$ is symmetric, which in turn is used in proving that $\phi^{-1}$ is Lipschitz, and is necessary for $\langle\cdot,\mathbf{H}_z \cdot\rangle$ to be a well-defined Riemannian metric.
>
> * While the authors did make assumptions and the methodology does have limitations, the authors did not address the consequences of these limitations in terms of the potential applications.
>
> There are a number of limitations to the methodology which are treated implicitly in various sections of the paper, but which we could bring together in a more detailed conclusion.
>
>
> 1) The bound in eq (2) indicates that $n$ should be "large" in order for $\mathbf{Q} \hat{X}_i$ to approximate $\phi_p(Z_i)$ well. $n$ also has an impact on the performance of Isomap: heuristically one needs a high density and a large number of points on or near the manifold to get a good estimate of the geodesic distance. Thus the methods we propose are likely to perform poorly when $n$ is small, corresponding e.g. to a graph with a small number of vertices. Indeed the impact of $n$ on reconstruction error is illustrated in the simulated data example, but as per our response to reviewer 5XMg, we shall produce additional results to help quantify errors and compare methods.
>
> 2) On the other hand, in applications involving large networks or matrices (e.g. cyber-security, recommender systems), the data encountered are often sparse. This is good news for computing the factorisation (in terms of speed), but bad news for statistical precision. In particular, for a graph, Equation 2 can only be expected to hold under logarithmically growing degree, the information-theoretic limit below which no algorithm can obtain asymptotically exact embeddings [1]. What this means in practice (as illustrated in our numerical results for the simulated data example in the appendix) is that the manifold may be very hard to distinguish, even in a large graph. There could be substantial estimation gains in better integrating the factorisation and manifold estimation steps.
>
> 3) As discussed above, our theory restricts attention to the case of positive-definite kernels. When $\mathbf{A}$ is say a correlation or covariance matrix the positive-definite assumption on the kernel is of course natural, but e.g. when $\mathbf{A}$ is an adjacency matrix, it is a less natural assumption. We are still in the process of understanding what happens when the positive-definite assumption is removed.
>
> [1] Abbe, Emmanuel. "Community detection and stochastic block models: recent developments." The Journal of Machine Learning Research 18.1 (2017): 6446-6531.

---

### Official Review · Reviewer_VpEa · 2021-07-16

**Rating:** 7
**Confidence:** 3

**Summary:**

This paper considers the problem of finding the latent variables $Z$ given noisy observations of $f(Z_i, Z_j)$ with unknown $f$, and suggests applying matrix factorization (e.g. spectral embedding) + nonlinear dimension reduction (e.g. isomap). It shows mathematically that under some assumptions on the kernel $f$, finding geodesic distance in the manifold (which the matrix factorization step approximates) is equivalent to finding the geodesic distance in the original latent space. Experiments on simulated and real-world data are provided.

**Limitations And Societal Impact:**

Yes.

**Main Review:**

This paper seems sufficiently different from the cited works [12] and [57], and appears to be the first to provide theoretical analysis with general kernel $f$ and continuous variables $Z$. This work could be an important step to more rigorously understanding why nonlinear dimensionality reduction methods work so well in practice.

The main question I have is regarding the air traffic data experiment. Why should LA and NY be far from each other in the embedding space? Is the hypothesis that the latent variable $Z_i$ is the geographical location of city $i$? Since there are a lot of other factors (e.g. population, economic size, airport size) that go into deciding whether there should be a flight between two cities, one could even argue that LA and NY as two coastal hubs should be close to each other in an embedding space. Also, how are we meant to interpret the structures in Figure 3 c? For example, what does it mean for points to be spread out vs. clumped? What are the clusters within North America?

This paper is written clearly and easy to follow, but I have a few clarification questions:
- Typo in Theorem 1, "Let $a$, $b$ any two points" -> "Let $a$, $b$ be any two points".
- In page 2 last sentence, a graph adjacency matrix is not necessarily sparse.
- Is the norm in (2) and the following paragraph the l2 norm?



**Time Spent Reviewing:**

3

---

> ### Author Response · Authors · 2021-08-10
> **Response to reviewer VpEa.**
>
> * Why should LA and NY be far from each other in the embedding space? Is the hypothesis that the latent variable $Z_i$ is the geographical location of city $i$? Since there are a lot of other factors (e.g. population, economic size, airport size) that go into deciding whether there should be a flight between two cities, one could even argue that LA and NY as two coastal hubs should be close to each other in an embedding space.
>
> In retrospect, this section lacks a lot of detail. The loose hypothesis is indeed that $Z_i$ is the geographical location of airport $i$, and that connections occur as a function of geodesic distance on Earth (a case covered by Proposition 2, line 208, treating the Earth as a sphere of radius one in some unit). But to make this hypothesis plausible, we apply degree-correction, through spherical projection of the embedding (as mentioned on line 266), to remove effects which indiscriminately magnify the probability of a connection from a given airport to *any* other, or the "airport's popularity". One could plausibly claim that the factors given by the reviewer --- population size, economic size, airport size --- only influence connectivity in this way.
>
> Our hope was that, after accounting for airport popularity, the probability of a connection would be sufficiently driven by geographic distance for some aspects of world geography to be recoverable through matrix factorisation followed by nonlinear dimension reduction into 2D.
>
> Indeed, Figures 3b),7a),7b),7d) (matrix factorisation followed by nonlinear dimension reduction) reveal true geographic structure not visible in Figures 3a),7c) (direct 2D matrix factorisation). Possibly because of the size of the Pacific Ocean, we do not see much improvement when we try to place the airports onto a sphere.
>
> We will make the hypothesis, and the importance of degree-correction, clearer in the revision. (There are arguments and empirical evidence that node2vec has degree-correction 'built-in' [1], so in those experiments we apply Isomap/umap/t-SNE to the embedding directly.)
>
> By contrast, if we somehow measured how "coastal" a city was, and computed the Euclidean distance between corresponding airports in that measure (so New York and LA would be close, Kansas City and LA would be far), we would not *a priori* expect this type of distance to have much bearing on the probability of a connection, especially after we correct for airport popularity (e.g. making it irrelevant whether coastal cities are more popular tourist destinations).
>
> * Also, how are we meant to interpret the structures in Figure 3 c? For example, what does it mean for points to be spread out vs. clumped? What are the clusters within North America?
>
> The clusters would ideally represent actual geographical clusters --- groups of airports in close proximity of each other. We have not yet had time to investigate this fully but, for example, in Figure 3c, the cluster on the extreme left comprises 203 airports of which 200 are in California, and their true geographical positions have an approximate interquartile range of 2 degrees in latitude and in longitude (about 200km in each direction). Similarly, when points are far apart, we would hope that the corresponding airports were geographically distant. At present we do not provide an explanation for any perceived variation in "clumping" across the graphs --- we are not sure such observations are significant (in fact the embeddings could be argued to be remarkably stable considering the complexity of our analysis) --- apart from for the month of the pandemic (April 20) where we did try to offer an explanation (in the supplementary material). In the revision we will include a few further experiments (to be determined) to measure the extent to which true geography is recovered, at least, for the case of North America.
>
> [1] Zhang, Yichi, and Minh Tang. "Consistency of random-walk based network embedding algorithms." arXiv preprint arXiv:2101.07354 (2021).
>
> * Typo in Theorem 1, "Let ,  any two points" -> "Let ,  be any two points".
>
> Thanks.
>
> * In page 2 last sentence, a graph adjacency matrix is not necessarily sparse.
>
> We did not mean to suggest it was; will simply remove "(when $\mathbf{A}$ is a graph adjacency matrix)".
>
> * Is the norm in (2) and the following paragraph the l2 norm?
>
> Yes, will clarify with "in Euclidean norm". (We reserve the subscript 2 for the infinite dimensional version, in e.g. line 128).

---

> > ### Comment · Reviewer_VpEa · 2021-09-10
> > **Acknowledgement**
> >
> > Thank you for your thoughtful response.

---

### Official Review · Reviewer_5XMg · 2021-07-16

**Rating:** 7
**Confidence:** 4

**Summary:**

The manuscript shows that by combining matrix factorisation with dimension reductions, one is able to reconstruct data (up to some invariances) by only considering a kernelised set. The main contribution is to show an explicit relation between the geodesic distances in the manifold that supports the kernelised data, and the distance of the original data set with respect to a Riemannian metric.

**Limitations And Societal Impact:**

These results are supplemented by experiments on both synthetic data, and two real datasets. The experiments are not incredibly in depth and it is hard to say exactly how "good" they are. I can appreciate the lack of benchmarks here, but if possible it would be nice to see some sort of comparison or discussion of the quality of the results.

**Main Review:**

The results themselves are interesting and open up possibilities for further research and applications. As this is a mainly theoretical paper, the experiments are not as important as the theoretical results and does not detract that much from the quality of the paper.

**Time Spent Reviewing:**

5

---

> ### Author Response · Authors · 2021-08-10
> **Response to reviewer 5XMg.**
>
> * These results are supplemented by experiments on both synthetic data, and two real datasets. The experiments are not incredibly in depth and it is hard to say exactly how "good" they are. I can appreciate the lack of benchmarks here, but if possible it would be nice to see some sort of comparison or discussion of the quality of the results.
>
> As the reviewer observes, benchmarks are difficult for real data examples because the kernel is unknown. (Small quantitative improvements for the flight network example are proposed in the response to reviewer VpEa.)  A fully quantitative comparison of methods will be undertaken for the simulated data example, where we have access to the latent positions and the kernel. On reflection, this would not only have aided comparison between methods but also a comparison of sparse and dense recovery. In the appendix, supplementary figures will be provided to report numerical values of the latent position recovery error (averaged over simulations). Firstly, a figure will be added to compare the different methods used in Figure 6, secondly, a figure will be added to provide quantitative results to follow on from the statements "as the theory predicts, the recovery error vanishes as $n$ increases" and "recovery error still shrinks, but more slowly" found on page 7, demonstrating the recovery error decreasing when increasing the number of points in the dense and sparse regimes.

---

### Official Review · Reviewer_MXyX · 2021-07-18

**Rating:** 6
**Confidence:** 3

**Summary:**

This paper considers the problem of sensing a notion of the true distance between nodes of a
network. This allows to estimate the true position of these nodes up to pairwise scaling. The
cradle of this approach is essentially organized into two steps: a) compute a matrix factorization
of the matrix associated with the graph, i.e., spectral embedding, and b) perform a nonlinear
dimensionality reduction step. The main observation, subject to necessary conditions, is that
Euclidean distances of the spectral embeddings agree with geodesic distances between the
graph nodes. Numerical experiments provide evidence that the proposed approach works.


**Limitations And Societal Impact:**

Yes

**Main Review:**

-) The paper is relatively well-written, although some improvements in written English would
be welcome (also, the authors use the word "between" too often).

-) The theory appears to be correct; at least up to the part I could confidently follow it. Moreover,
most concepts discussed in the paper are clear, and previous work is mentioned.

-) One think I missed is comparison with other methods. Is the case that no other methods
can solve these problems? Euclidean distance to geodesic distance mappings are a well
studied topic.

-) Other applications can be found in:
https://www.isca-speech.org/archive/archive_papers/interspeech_2010/i10_2742.pdf
https://arxiv.org/pdf/1902.01395.pdf

What is new compared to those viewpoints?

-) It is my impression that the new approach need a few parameters to have the proper
values in order to work. How the value of hyperparameters (e.g., dimension of spectral
embeddings) affects the performance of the proposed approach for the real datasets?
How about other methods?

-) Line 109: did you mean to say "Wasserstein distance"?

-) Line 112: "legitimately" is a rather strange word in this context. Please rephrase.

**Time Spent Reviewing:**

4

---

> ### Author Response · Authors · 2021-08-10
> **Response to reviewer MXyX.**
>
> * The paper is relatively well-written, although some improvements in written English would be welcome (also, the authors use the word "between" too often).
>
> Thanks for pointing this out, we'll see if we can reduce repetition in the revision.
>
> * One think I missed is comparison with other methods. Is the case that no other methods can solve these problems? Euclidean distance to geodesic distance mappings are a well studied topic.
>
> We should have made it clearer that we are aware of *no other approach* to (matrix factorisation + nonlinear dimension reduction) to recover the $Z_i$'s given a graph or similarity matrix which does not require the user to specify anything about the kernel $f$. We will make this explicit in the revision.
> Of course, there are many methods to recover the $Z_i$'s when the kernel is *known* (and we review those possibilities from line 36). In view of the reviewer's comment, we will include a short demonstration that the "known kernel" approach does not work if the kernel is wrong (effectively always the case in real data).
>
> However, within the class of methods which combine matrix factorisation + nonlinear dimension reduction in some form or another, we *do* make comparisons (either spectral embedding or node2vec for the factorisation, and either Isomap, t-SNE, UMAP for the nonlinear dimension reduction). These comparisons are reported in the appendix. In the revision, this comparison will include more quantitative experiments in line with reviewer 5XMg's comments.
>
> * Other applications can be found in: https://www.isca-speech.org/archive/archive_papers/interspeech_2010/i10_2742.pdf https://arxiv.org/pdf/1902.01395.pdf . What is new compared to those viewpoints?
>
> Thanks very much for bringing these references to our attention.
>
> Regarding the Karam and Campbell paper, there are some methodological and terminological differences between their paper and ours which need to be considered carefully to make a detailed comparison. For example, they start from feature vectors and then calculate the inner-products and Euclidean distances between them,  whereas we obtain feature vectors by factorisation (e.g. spectral decomposition) of an observed matrix; and they use "graph embedding" to refer to the subsequent construction of an $\epsilon$- or nearest neighbor graph, whereas we follow the convention that "spectral embedding" of a graph refers to obtaining feature vectors from spectral decomposition of, e.g., its adjacency matrix in the first place.
>
> However, stepping back from these details, the most significant difference is that the Karam and Campbell paper does not provide any rigorous mathematical justification for their procedure, or explanation of why manifold structure may be present in the first place. By contrast, it is our main contribution to, for the first time, give a fully rigorous explanation of *why* finding geodesic distance can recover latent positions.
>
> The Karam and Campbell paper is just one example of how, in general terms, the combination of matrix factorisation (or graph embedding, PCA, classical multi-dimensional scaling, and so forth) followed by nonlinear dimension reduction (e.g. Isomap, t-SNE, UMAP) has pervaded many areas of science, but without a rigorous justification or unifying perspective. The reviewer's comment has helped us understand that we should signpost such applications in order to clarify our contribution --- to provide this justification --- and we will do so in the revision. Other example applications which we could refer to include: genetics [1], Natural Language Processing [2], cyber-security [3].
>
> The Yamin et al. paper is somewhat further removed from our work. They consider positive-definite matrix-valued data and consider geodesic distance on the manifold of such matrices. The manifolds we consider are completely different and arise from the image of the feature map of positive-definite kernels (that there happens to be positive-definiteness involved in both cases is just a coincidence). This difference makes close comparisons not very meaningful, although one broad comparison is that (as in the Karam and Campbell) Yamin et al. don't provide rigorous justification for using geodesic distance, in contrast to our contribution.
>
> [1] Margaryan, Ashot, et al. "Population genomics of the Viking world." Nature 585.7825 (2020): 390-396.
>
> [2] Hasan, Souleiman, and Edward Curry. "Word re-embedding via manifold dimensionality retention." Association for Computational Linguistics (ACL), 2017.
>
> [3] Anglade, Thomas, Christophe Denis, and Thierry Berthier. "A novel embedding-based framework improving the User and Entity Behavior Analysis." https://hal.archives-ouvertes.fr/hal-02316303
>
> * It is my impression that the new approach need a few parameters to have the proper values in order to work. How the value of hyperparameters (e.g., dimension of spectral embeddings) affects the performance of the proposed approach for the real datasets? How about other methods?
>
> The reviewer will remember that we recommended the method of Zhu & Ghodsi (2006) for dimension selection on line 106. When we apply this to the graphs of Section 4.2, the estimated dimension varies across graphs, especially going into the pandemic (the April 20 graph), but it is always below 10. Since we wish to choose a single, common, dimension to allow a robust comparison between the graphs, we adopt the recommendation of [4] to overestimate $p$ (whence our choice $p=10$), at the potential expense of added variance, to be sure to capture all the signal. In the revision, we will explain this choice, and include analogous visualisations for a range of different $p$ in the supplementary material. When $p$ is much higher, we expect the pictures to be similar but noisier, whereas when it is much lower we expect the pictures to be materially different, because of a loss of signal. We will also vary $p$ in the experiments using node2vec. There are several other hyper-parameters at play in node2vec and the different non-linear dimension reduction techniques we compare (e.g. random walk parameters (node2vec), perplexity (t-SNE), number of neighbours (UMAP), minimum distance (UMAP)) but for those we just use the default choices and we will make this explicit.
>
> [4] Athreya, Avanti, et al. "Statistical inference on random dot product graphs: a survey." The Journal of Machine Learning Research 18.1 (2017): 8393-8484.
>
> * Line 109: did you mean to say "Wasserstein distance"?
>
> Yes, thanks.
>
> * Line 112: "legitimately" is a rather strange word in this context. Please rephrase.
>
> What we meant is that, while the term "manifold" is sometimes used quite loosely in the ML literature, we actually rigorously establish that $\mathcal{M}$ is a topological manifold in the conventional mathematical sense. The word "legitimate" was supposed to make clear that we were conforming with precise mathematical definitions. We will replace it with "can be called a topological manifold in the conventional mathematical sense".

---

### Author Response · Authors · 2021-08-10
**Thanks**

We would like to thank all the reviewers for their time and expertise. We have responded to each review separately.

---

### Decision · Program_Chairs · 2021-09-27

**Decision:**

Accept (Poster)

**Comment:**

The paper revisits the problem of recovering positions of points given the similarity information. It proposes to do: first, a spectral embedding and second, a nor linear embedding to obtain the positions. As reviewers note, the interesting contributions are in connecting the geometry between spectral approximations (X) and true embeddings (Z). Overall, the theory and experiments are interesting. A minor comment, currently, the paper assumes that A is fully known and the theory goes through. What happens A is partially known (akin to recovering positions from partial similarity information). What kinds of consistency results are required for the approximations to succeed?